# HLA alleles, disease severity, and age associate with T-cell responses following infection with SARS-CoV-2

Thorunn A. Olafsdottir [1,9 ✉], Kristbjorg Bjarnadottir[1,9], Gudmundur L. Norddahl[1], Gisli H. Halldorsson [1], Pall Melsted[1,2], Kristbjorg Gunnarsdottir[1], Erna Ivarsdottir [1], Thorhildur Olafsdottir [1], Asgeir O. Arnthorsson[1], Fannar Theodors[1], Elias Eythorsson[3], Dadi Helgason[3], Hannes P. Eggertsson [1], Gisli Masson[1], Sólveig Bjarnadottir[3,4], Saedis Saevarsdottir [1,4,5], Hrafnhildur L. Runolfsdottir [3], Isleifur Olafsson[6], Jona Saemundsdottir[1], Martin I. Sigurdsson[4,7], Ragnar F. Ingvarsson[3], Runolfur Palsson [3,4], Gudmundur Thorgeirsson[1,4], Bjarni V. Halldorsson [1,8], Hilma Holm [1], Mar Kristjansson[3], Patrick Sulem [1], Unnur Thorsteinsdottir[1,4], Ingileif Jonsdottir [1,4], Daniel F. Gudbjartsson [1,2] & Kari Stefansson [1,4 ✉]

Memory T-cell responses following SARS-CoV-2 infection have been extensively investigated but many studies have been small with a limited range of disease severity. Here we analyze SARS-CoV-2 reactive T-cell responses in 768 convalescent SARS-CoV-2-infected (cases) and 500 uninfected (controls) Icelanders. The T-cell responses are stable three to eight months after SARS-CoV-2 infection, irrespective of disease severity and even those with the mildest symptoms induce broad and persistent T-cell responses. Robust CD4+ T-cell responses are detected against all measured proteins (M, N, S and S1) while the N protein induces strongest CD8+ T-cell responses. CD4+ T-cell responses correlate with disease severity, humoral responses and age, whereas CD8+ T-cell responses correlate with age and functional antibodies. Further, CD8+ T-cell responses associate with several class I HLA alleles. Our results, provide new insight into HLA restriction of CD8+ T-cell immunity and other factors contributing to heterogeneity of T-cell responses following SARS-CoV-2 infection.

[1] deCODE genetics/Amgen Inc., Reykjavik, Iceland. [2] School of Engineering and Natural Sciences, University of Iceland, Reykjavik, Iceland. [3] Internal Medicine and Emergency Services, Landspitali – The National University Hospital of Iceland, Reykjavik, Iceland. [4] Faculty of Medicine, School of Health Sciences, University of Iceland, Reykjavik, Iceland. [5] Department of Medicine, Landspitali, The National University Hospital of Iceland, Reykjavik, Iceland. [6] Clinical Laboratory Services, Diagnostics and Blood Bank, Landspitali – The National University Hospital of Iceland, Reykjavik, Iceland. [7] Perioperative Services, Landspitali – The National University Hospital of Iceland, Reykjavik, Iceland. [8] School of Science and Engineering, Reykjavik University, Reykjavík, Iceland. [9] These authors contributed equally: Thorunn A. Olafsdottir, Kristbjorg Bjarnadottir. ✉email: thorunn.olafsdottir@decode.is; kstefans@decode.is

Coronavirus disease 2019 (COVID-19) caused by severe acute respiratory coronavirus 2 (SARS-CoV-2) has a wide range of clinical manifestations from no or mild symptoms to severe disease, with risk of respiratory failure and death, especially in the elderly and those with serious preexisting conditions[1]. It is therefore important to understand the heterogeneity of the immune response induced by SARS-CoV-2 to inform better treatment and vaccination strategies.

Antibody measurements in plasma or serum are relatively easy to perform on a large scale and are therefore an important diagnostic tool to evaluate exposure to the virus as well as studying the breadth and kinetics of the immune response following infections. We reported on seroprevalence of SARS-CoV-2 infection in Iceland during the first COVID-19 wave[2], where we showed that 91.1% of individuals with a prior diagnosis of COVID-19 by qPCR had seroconverted and that SARS-CoV-2 antibody levels correlated positively with clinical severity and requirement for hospitalization. Neutralizing antibodies against the receptor binding domain (RBD) of the S1 subunit correlate with protection against COVID-19 as they prevent the binding of SARS-CoV-2 to human cells expressing angiotensin-converting enzyme 2 (ACE2) and subsequent entry into cells[3].

In addition to antibodies, antigen-specific T-cells play a central role in the immune response against viruses. CD8[+] cytotoxic T-cells eliminate virally infected cells and, as such, participate in reducing viral replication and mitigating disease severity, while CD4[+] T helper cells shape the overall immune response, including the antibody and CD8[+] T-cell responses, and this combined adaptive immune response mediates recovery and protects against reinfection[4]. Hence, persistent CD8[+] and CD4[+] T-cell memory is critical for long-term protection against COVID-19 and their role has recently been highlighted by the notion that SARS-CoV-2 T-cell responses are not substantially affected by the mutations found in the emerging SARS-CoV-2 variants[5]. In individuals infected with SARS-CoV-1, a coronavirus that caused an epidemic in Asia in 2003, T-cell responses were detected up to 17 years after infection, demonstrating the longevity of memory T-cells[6], whereas circulating antibody levels declined substantially within the first 2–3 years[7,8]. Several papers have reported robust T-cell responses to SARS-CoV-2 derived peptides following infection[9–14]. However, analyses of antigen-specific T-cell responses are more challenging to perform on a large scale than antibody responses and are therefore often underpowered to robustly study demographic and clinical correlates of T-cell responses e.g., age, sex, and disease severity[15]. Further, T-cell receptors recognize pathogenic peptides presented in the conformational structure of the antigen binding-groove of a human leukocyte antigen (HLA) molecule, emphasizing the importance of studying whether sequence polymorphisms at the HLA locus (HLA alleles) associate with T-cell responses following infection of SARS-CoV-2.

To address the need for well-powered T-cell analysis following SARS-CoV-2 infection, we measured T-cell responses in a large number of SARS-CoV-2-infected cases ($n = 768$) and uninfected controls ($n = 396$) from the first SARS-CoV-2 wave in Iceland. Due to extensive qPCR and serological screening in Iceland, as well as clinical monitoring of everyone with a qPCR confirmed SARS-CoV-2 diagnosis, our SARS-CoV-2 cases and uninfected controls were exceptionally well characterized. We studied functional anti-viral immunity by assessing CD4[+] and CD8[+] T-cells secreting the canonical type 1 cytokines: IFN-γ, TNF-α and IL-2 upon stimulation with SARS-CoV-2 proteins[16] in COVID-19 cases with a wide range of disease severity, collected up to eight months after diagnosis of infection, and compared them to uninfected controls. Furthermore, we studied the correlation of SARS-CoV-2 reactive T-cell responses with HLA alleles, age, sex, disease severity and humoral responses.

## Results

### Sex and age but not SARS-CoV-2 infection affected total T-cell count.
On average, we counted 68,666 CD4[+] T-cells and 32,428 CD8[+] T-cells per sample from 768 SARS-CoV-2 convalescent cases, 148 pre-pandemic controls and 396 uninfected controls collected during the pandemic (Table 1). CD4[+] T-cell counts were 10.7% (95% CI: 7.6%, 13.8%, $P = 1.5e-12$) higher among females than males but did not vary significantly with age, while CD8[+] T-cell counts were 4.1% lower among females than males (95% CI: 0.3%, 7.7%, $P = 0.030$), and decreased by 0.6% per year of age (95% CI: 0.5%, 0.8%, $P = 3.8e-22$, Supplementary Table 1). After accounting for age and sex, the pre-pandemic samples had, on average, 9.0% (95% CI: 4.5%, 13.2%, $P = 1.2e-4$) fewer CD4[+] T-cells than the samples collected during the pandemic, likely reflecting the negative effect of sample storage on the CD4[+] T-cell count[17]. We saw no significant difference in CD8[+] T-cell counts between samples collected before and during the pandemic. There was no difference in total CD4[+] or CD8[+] T-cell counts between cases or controls after accounting for age and sex (Supplementary Table 1).

### All four SARS-CoV-2 proteins induced CD4[+] T-cell responses in cases.
We studied SARS-CoV-2 T-cell responses against four structural SARS-CoV-2 proteins (M, N, S, and S1) and analyzed type 1 cytokine responses by measuring all IFN-γ[+], TNF-α[+], or IL-2[+] producing T-cells that were further split into cells producing one, two, or all three cytokines (polyfunctional IFN-γ[+]TNF-α[+]IL-2[+] T-cells). We performed linear regression to test for associations between T-cell responses in different groups and report effects that represent differences in the logarithm of T-cell responses, either as the frequency of cytokine secreting cells out of all CD4[+] T-cells or the absolute count of CD4[+] cytokine secreting T-cells (Supplementary Table 2 and corresponding information for CD8[+] T-cell responses in Supplementary Table 3). For simplification, we only refer to the effects on the frequency of T-cell responses in the text and give the effect range (ranging from the lowest to highest effect), when referring to multiple cell sub-populations and/or different stimulations.

We detected significant CD4[+] T-cell responses in cases against all four proteins compared to unstimulated cultures (Fig. 1 and Supplementary Fig. 1). CD4[+] T-cells from controls responded to stimulation with the M and S proteins, but not to N and S1 stimulations, indicating cross-reaction with components of other common human coronaviruses (HCoVs) that share epitopes with two out of the four proteins tested (Fig. 1c and Supplementary Fig. 1).

The frequencies of all IFN-γ[+], TNF-α[+] and IL-2[+] SARS-CoV-2 reactive CD4[+] T-cells were higher (effect range 0.41 to 1.45) among cases than controls for all four proteins (Supplementary Table 2). These SARS-CoV-2 reactive CD4[+] T-cells were polyfunctional (effect range 0.58 to 0.99), double cytokine-producing, either IFN-γ[+]TNF-α[+] (effect range 0.22 to 0.51) or TNF-α[+]IL-2[+] (effect range 1.01 to 1.56) and single-positive TNF-α[+] (effect range 0.25 to 0.43) or IL-2[+] (effect range 0.15–0.24) cells. However, IFNγ[+] single-positive cells were only increased upon N protein stimulation (effect:0.06, 95% CI: 0.04,0.08, $P = 1e-7$) (Fig. 1d and Supplementary Table 2). We detected very few double-positive cytokine-producing IFN-γ[+]IL-2[+] SARS-CoV-2 reactive CD4[+] T-cells and their frequency was not significantly higher in cases than controls after accounting for multiple testing ($P > 0.05/60 = 8.3e-4$, Supplementary Fig. 1, and Supplementary Table 2).

We estimated the correlation between T-cell responses with Spearman's correlation coefficient (ρ). Overall, the SARS-CoV-2 induced responses of polyfunctional and double-positive TNF-α[+]IFN-γ[+], as well as TNF-α[+]IL-2[+] CD4+ T-cells, correlated strongly between the different SARS-CoV-2 proteins among cases, demonstrating that CD4[+] T-cell responses in

**Table 1 Characteristics of persons contributing blood samples for the study.**

|  | SARS-CoV-2 cases | Pre-pandemic controls | Controls collected during pandemic |
|---|---|---|---|
| Individuals, N | 768 | 148 | 396 |
| Blood samples, N | 863 | 148 | 403 |
| Age at sampling in years (SD) | 43.8 (15.3) | 55.5 (11.2) | 54.8 (15.0) |
| Female, N (%) | 411 (53.5%) | 87 (58.8%) | 237 (59.8%) |
| Sample collection dates | May–December 2020 | October 2001–February 2020 | June–December 2020 |
| Pre-pandemic samples, N (%) | 80 (10.4%) | 148 (100%) | 44 (11.1%) |
| Median time from diagnosis, days (first and third quartiles) | 124 (105, 212) | −1038 (−1412, −282) | NA |
| Longitudinal samples, N (%) | 94 (12.2%) | NA | 7 (1.8%) |
| COVID-19 severity |  |  |  |
| Mild | 209 | NA | NA |
| Moderate | 140 | NA | NA |
| Severe | 167 | NA | NA |
| Hospitalized | 30 | NA | NA |
| Not available | 222 | NA | NA |
| Diagnosis of SARS-CoV-2 infection |  |  |  |
| qPCR-positive, N (%) | 689 (89.7%) | NA | 22 (5.6%) |
| Ever pan-Ig anti-N antibody positive, N (%)* | 757 (98.6%) | NA | 0 (0%) |
| Ever pan-Ig anti-RBD antibody positive, N (%)** | 763 (99.3%) | NA | 0 (0%) |
| Ever pan-Ig anti-RBD antibody positive, N (%)*** | 763 (99.3%) | NA | 0 (0%) |
| No pan-antibody assay positive, N (%) | 0 (0%) | NA | 396 (100%) |
| One pan-antibody assay positive, N (%) | 0 (0%) | NA | 0 (0%) |
| Two pan-antibody assay positive, N (%) | 21 (2.7%) | NA | 0 (0%) |
| Three pan-antibody assay positive, N (%) | 747 (97.2%) | NA | 0 (0%) |
| T-cell counts |  |  |  |
| Passing CD4 + T-cell count criteria, N (N samples) | 767 (862) | 148 (148) | 392 (399) |
| Mean CD4 + T-cell count among passing criteria (SD) | 69377 (17173) | 64903 (18931) | 69246 (18705) |
| Passing CD8 + T-cell count criteria, N (N samples) | 764 (859) | 148 (148) | 387 (394) |
| Mean CD8 + T-cell count among passing criteria (SD) | 33398 (11386) | 30148 (11111) | 31889 (13343) |

*Roche.
**Wantai.
***Roche.

cases were not restricted to any of these SARS-CoV-2 proteins (Supplementary Figs. 2–4). However, only S and M proteins stimulated CD4+ T-cell responses among controls, reflected in the correlation between the two stimulation settings in polyfunctional (Spearman's ρ = 0.61, 95% CI: 0.52, 0.69) and double-positive TNF-α+IFN-γ+ (Spearman's ρ = 0.80, 95% CI: 0.75, 0.86) CD4+ T-cells, consistent with the notion that responses induced by those proteins in SARS-CoV-2 uninfected individuals reflect cross-reactive T-cell responses and not false positive responses (Supplementary Fig. 2). Further, we note that there is a low but significant correlation between T-cell responses against each of the SARS-CoV-2 and the CMV peptides indicating that there is an individual difference in memory T-cell responses among the participants in the study (Supplementary Figs. 2–4).

**IFN-γ secreting CD8+ T-cell responses were most strongly induced against the N protein.** Cases had higher frequencies than controls of IFN-γ+ CD8+ T-cells responding to all four proteins as estimated by linear regression (effect range 0.21 to 1.07, Fig. 2 and Supplementary Table 3). These IFN-γ+ CD8+ T-cells were mainly double-positive IFN-γ+TNF-α+ (effect range 0.16 to 0.81), but also polyfunctional (effect range 0.05 to 0.30) and single-positive (effect range 0.05 to 0.40). The strongest SARS-CoV-2 reactive CD8+ T-cell responses were induced by the N protein: all IFN-γ+ (effect = 1.07, 95% CI: 0.95, 1.21, P = 1.7e-50), and double-positive IFN-γ+TNF-α+ (effect = 0.81, 95% CI: 0.69, 0.93, P = 4.4e-38; Fig. 2c, d and Supplementary Table 3). There were no significant effects on TNF-α+ or IL-2+ CD8+ T-cells that did not simultaneously produce IFN-γ+ following the

SARS-CoV-2 protein stimulation (Supplementary Table 3). There was little SARS-CoV-2 reactivity in CD8+ T-cells from controls, suggesting lower cross-reactivity of CD8+ T-cells with HCoV epitopes than of CD4+ T-cells (Fig. 2a, c).

**HLA alleles associated with SARS-CoV-2 reactive T-cell responses.** Given the fundamental role of the antigen binding-groove conformational structure in the HLA molecule's capacity to present different pathogenic peptides to T-cells, we tested the association between T-cell responses and all four-digit HLA alleles for class I and II genes, as well as HLA-E, HLA-F and HLA-G. We had HLA genotypes for 742 out of the 768 cases in the study. We used linear regression to test for the association of T-cell responses with the HLA genotype and reported the effects as changes in the logarithm of T-cell responses. The strongest association was between increasing IFN-γ secreting N-reactive CD8+ T-cell responses and HLA-B*07:02 (effect = 1.08, 95% CI: 0.93, 1.22, P = 7.9e-46) followed by HLA-C*07:02 (effect = 0.94, 95% CI:0.80, 1.08, P = 1.0e-37). The association between N-reactive CD8+ T-cell responses and HLA-C*07:02 as well as other HLA alleles (meeting the multiple testing P value cutoff P < 4.8e-8) were explained by HLA-B*07:02 (Table 2 and Supplementary Table 4). S1-reactive CD8+ T-cell responses also associated strongly with HLA-C*07:02 (effect = 0.46, 95% CI: 0.33, 0.58, P = 3.0e-12) and HLA-B*07:02 (effect = 0.47, 95% CI: 0.33, 0.59, 6.2e-12) with HLA-C*07:02 having slightly more significant P value and therefore chosen as the lead signal. However, given the high linkage disequilibrium between them, it is impossible to know which one represents the causative allele. Independent of HLA-C*07:02, HLA-A*01:01 associated with

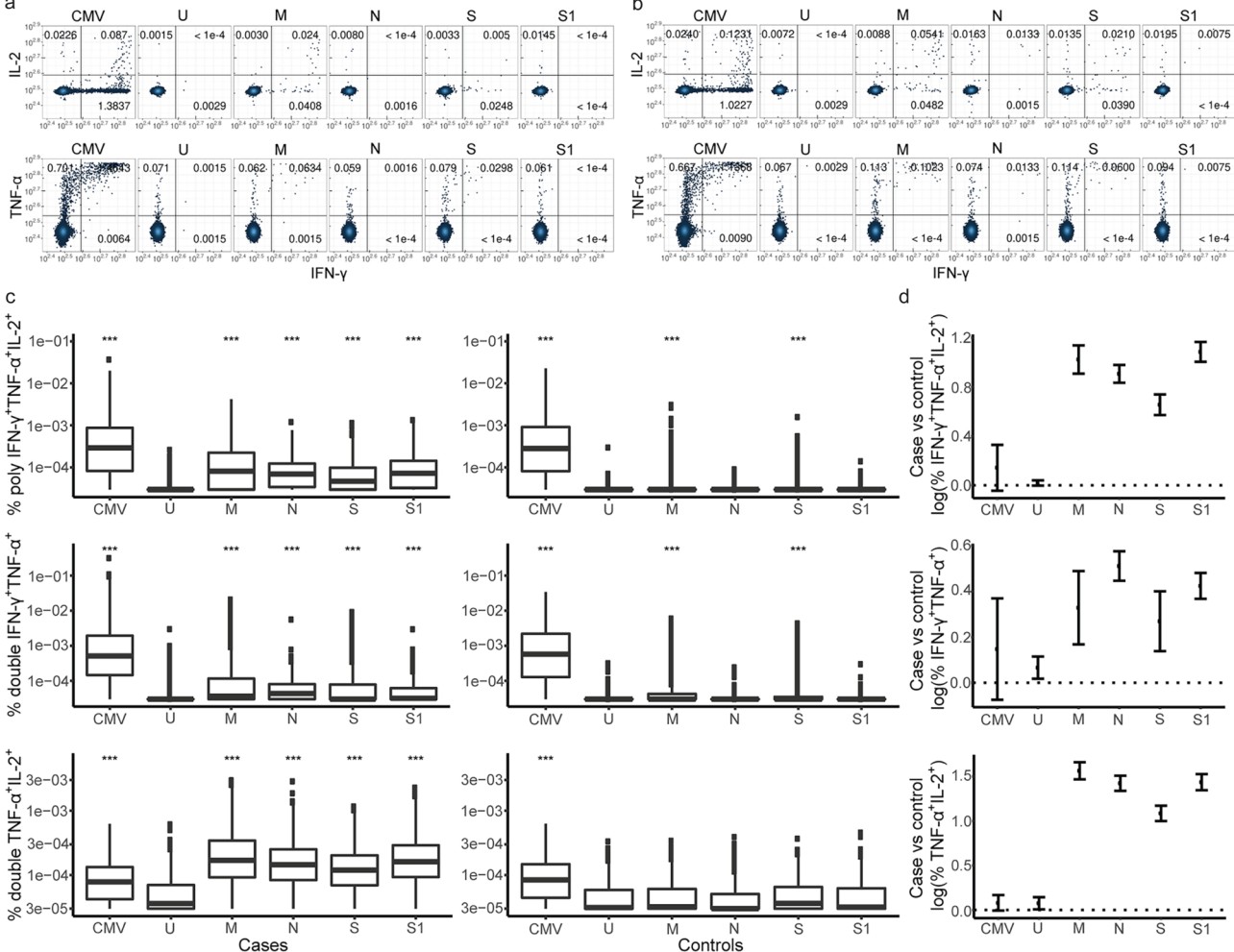

**Fig. 1 Polyfunctional CD4$^+$ T-cell responses are induced against all four SARS-CoV-2 proteins.** Representative flow cytometry plots showing the cytokine response of SARS-CoV-2 reactive CD4$^+$ T-cells from **a** an uninfected control and **b** a SARS-CoV-2 case for all stimulation conditions. The **a** uninfected control (sampled before March 2020) and **b** case (sampled after March 2020) samples come from the same individual. **c** Box plots show the frequencies of each CD4$^+$ T-cell functional phenotype responding to the different stimulation conditions in SARS-CoV-2 cases ($n = 767$) and uninfected controls collected during the pandemic ($n = 392$). Data for polyfunctional (IFN-$\gamma^+$TNF-$\alpha^+$IL-2$^+$) and double-positive (IFN-$\gamma^+$TNF-$\alpha^+$ and TNF-$\alpha$ + IL-2$^+$) CD4$^+$ T-cells are shown. The bottom and top of the boxes correspond to the 25th (Q1) and 75th (Q3) percentiles, the line inside the box corresponds to the median, and the whiskers are located at max(min(Expression), Q1 – 1.5 IQR) and min(max(Expression), Q3 + 1.5 IQR), respectively (where IQR is the interquartile range = Q3 – Q1). ***$P < 0.001$ for indicated stimulation condition compared with unstimulated culture (U). **d** The plot shows estimates for the difference in the logarithm of the percentage of polyfunctional, double-positive IFN-$\gamma^+$TNF-$\alpha^+$ as well as TNF-$\alpha^+$IL-2$^+$ CD4$^+$ T-cells within the total CD4$^+$ T-cell population responding to the different stimulation conditions between cases ($n = 767$) and uninfected controls collected during the pandemic ($n = 392$). The error bars indicate 95% confidence intervals (CI). The different stimulation conditions are a positive control (CMV), the unstimulated control (U), and the four different SARS-CoV-2 proteins (M, N, S, and S1). See also Supplementary Fig. 1 and Supplementary Table 2. Association between T-cell responses in different groups was tested by linear regression and significance was assessed with the standard linear regression t-test.

decreasing S1-reactive CD8$^+$ T-cell responses (effect$_{adjusted}$ = −0.42, 95% CI: −0.58, −0.27, $P_{adjusted}$ = 1.5e-7). Interestingly, the only association observed with S-reactive CD8$^+$ T-cell responses was with the non-classical HLA class I allele HLA-G*01:04 (effect = 0.51, 95% CI: 0.38, 0.65, $P$ = 7.5e-14) and no significant association was observed between HLA alleles and M-reactive CD8$^+$ T-cell responses. Increasing CMV-reactive CD8$^+$ T-cell responses associated with HLA-C*07:02 (effect = 0.82, 95% CI: 0.53, 1.11, $P$ = 3.8e-8) and HLA-B*07:02 (effect = 0.84, 95% CI: 0.54, 1.14, $P$ = 5.8e-8), again HLA-C*07:02 is defined as the lead signal based on the slightly better $P$ value. The only association we observed between HLA alleles and CD4$^+$ T-cell responses was between higher M-reactive poly-functional CD4$^+$ T-cell responses and HLA-DQA1*05:05 (effect = 0.59, 95% CI: 0.39, 0.80, $P$ = 1.4e-8; Table 2). No

significant association was observed between T-cell associating HLA alleles and disease severity after correcting for multiple testing (Supplementary Table 5). In addition to testing the association between T-cell responses and HLA alleles, we searched for associations across the genome, but no sequence variants outside the HLA region reached genome-wide significance.

Using netMHCpan v4.0[18] to predict peptide binding across the associating HLA alleles, we found that all of the classical class I HLA alleles associating with SARS-CoV-2 reactive CD8$^+$ T-cell responses are predicted to bind a substantial number of peptides derived from the corresponding protein (Table 2 and Supplementary Data 1). Interestingly, one of the peptides binding to HLA-B*07:02, N$_{106-117}$ (PRWYFYYLGTGP), has been reported to be well conserved in two out of four common human coronaviruses (HCoV-OC43 and HCoV-HKU1)[19]. Further, many of the other N peptides predicted to

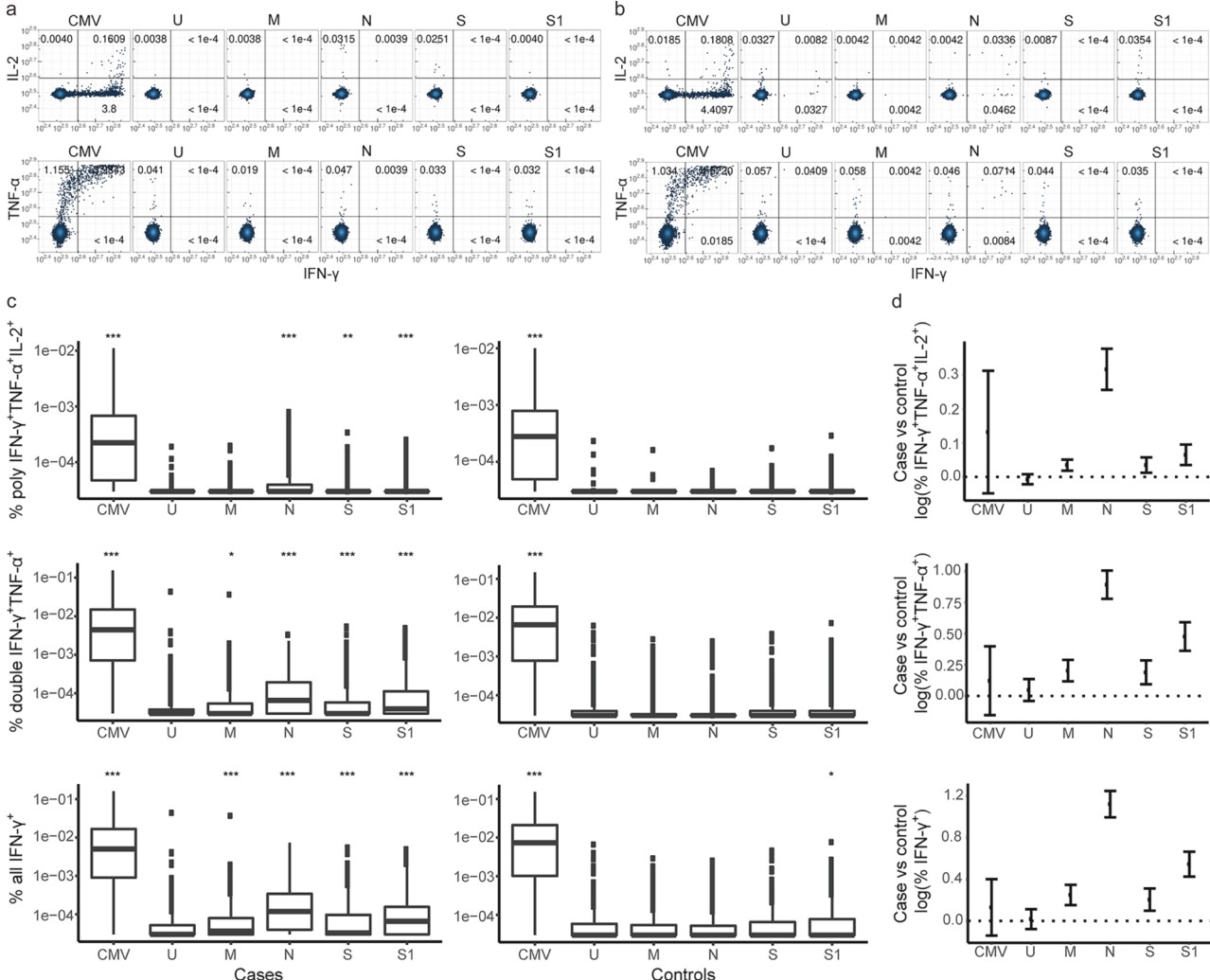

**Fig. 2 IFN-γ secreting CD8+ T-cell responses are most strongly induced against the N protein.** Representative flow cytometry plots showing the cytokine response of SARS-CoV-2 reactive CD8+ T-cells from **a** an uninfected control and **b** a SARS-CoV-2 case for all stimulation conditions. The **a** uninfected control (sampled before March 2020) and **b** case (sampled after March 2020) samples come from the same individual. **c** Box plots show the frequencies of each CD8+ T-cell functional phenotype responding to the different stimulation conditions in SARS-CoV-2 cases ($n = 764$) and uninfected controls collected during the pandemic ($n = 387$). Polyfunctional (IFN-γ+TNF-α+IL-2+) and double-positive (IFN-γ+TNF-α+), as well as all IFN-γ+ CD8+ T-cells, are shown. The bottom and top of the boxes correspond to the 25th (Q1) and 75th (Q3) percentiles, the line inside the box corresponds to the median, and the whiskers are located at max(min(Expression), Q1 – 1.5 IQR) and min(max(Expression), Q3 + 1.5 IQR), respectively (where IQR is the interquartile range = Q3 – Q1). *$P < 0.05$ and ***$P < 0.001$ for indicated stimulation condition compared with unstimulated culture (U). **d** The plots show estimates for the difference in the logarithm of the percentage of polyfunctional, double-positive IFN-γ+TNF-α+ as well as all INF-γ+ CD8+ T-cells within the total CD8+ T-cell population responding to the different stimulation conditions between cases ($n = 764$) and controls collected during the pandemic ($n = 387$). The error bars indicate 95% confidence intervals (CI). The different stimulation conditions are a positive control (CMV), unstimulated control (U), and the four different SARS-CoV-2 proteins (M, N, S, and S1). See also Supplementary Table 3. Association between T-cell responses in different groups was tested by linear regression and significance was assessed with the standard linear regression t-test.

be presented by HLA*B07:02 are located in regions of the N protein with high homology between SARS-CoV-2 and other *Coronaviridae* (Supplementary Data 1 and Supplementary Fig. 5a). The same was true for the peptides predicted to be presented by HLA alleles associating with S1-reactive CD8+ T-cell responses, although amino acid sequences of the S1 protein were more heterogeneous than of the N protein (Supplementary Data 1 and Supplementary Fig. 5).

**SARS-CoV-2 reactive CD4+ T-cell responses correlated with age, sex, and disease severity while CD8+ T-cell responses only correlated with age.** We analyzed the association between age and T-cell responses with linear regression by splitting cases and

controls into three age bins (18–40, 41–60, and 61–91 years of age) and report differences in the logarithm of T-cell responses, per age bin. SARS-CoV-2 reactive CD4+ T-cell responses increased with age in cases (effect range 0.55 to 2.37) but not in controls (Fig. 3a). N-reactive CD8+ T-cell responses increased somewhat with age (effect = 0.85, 95% CI: 0.30, 1,41) but not the responses to other SARS-CoV-2 protein stimulations (Fig. 3b). Among both cases and controls, CD4+ T-cell response to CMV stimulation increased until around 50 years of age and then stabilized or decreased slightly while CD8+ T-cell response to CMV increased slightly with age. There was no age effect in unstimulated cultures. After accounting for multiple testing, there was not a significant difference in SARS-CoV-2 or CMV-reactive

**Table 2 HLA alleles associated with CD8+ and CD4+ T-cell responses.**

| Protein | HLA allele | AF (%) | P value | Effect (95% CI) | P value_adj | Effect_adj (95% CI) | Covariate | Predicted no of peptides[a] | Strong binders[b] |
|---|---|---|---|---|---|---|---|---|---|
| **CD8+T-cell responses** | | | | | | | | | |
| N | B*07:02 | 18.3 | $7.9 \times 10^{-46}$ | 1.08 (0.93, 1.22) | - | - | - | 77 | 18 |
| S1 | C*07:02 | 20.6 | $3.0 \times 10^{-12}$ | 0.47 (0.33, 0.60) | - | - | - | 107 | 24 |
| S1 | A*01:01 | 10.7 | $3.8 \times 10^{-9}$ | -0.49 (-0.65, -0.33) | $1.5 \times 10^{-7}$ | -0.42 (-0.58, -0.27) | C*07:02 | 175 | 47 |
| S | G*01:04 | 11.0 | $7.5 \times 10^{-14}$ | 0.52 (0.38, 0.65) | - | - | - | - | - |
| CMV | C*07:02 | 20.6 | $3.7 \times 10^{-8}$ | 0.82 (0.53, 1.11) | - | - | - | 56 | 12 |
| **CD4 + T-cell responses** | | | | | | | | | |
| M | DQA1*05:05 | 6.0 | $1.4 \times 10^{-8}$ | 0.55 (0.36, 0.74) | - | - | - | | |

*AF* allele frequency, *P value*_adj *P* value adjusted for the lead associating allele, *Effect*_adj effect associated for the lead associating allele.
[a]netMHCpan v4.0[18] was used to predict the binding of SARS-CoV-2 and CMV-derived peptides with indicated HLA alleles.
[b]Strong binders were defined by netMHCpan v4.0 as peptides that rank in the top 0.5% of the predicted affinity compared to a set of random neutral peptides. Other peptides that are predicted to bind corresponding HLA allele rank between the top 0.5–2%. *P* value cutoff was determined as $4.8 \times 10^{-8}$, adjusting for the number of all HLA alleles. $n = 742$ HLA-typed SARS-CoV-2-infected cases.

CD4+ or CD8+ T-cell responses between males and females (Supplementary Table 6).

Assessment of severity of COVID-19 symptoms was available for 546 out of the 768 cases, as all Icelanders diagnosed with SARS-CoV-2 infection defined by qPCR were monitored by the telehealth monitoring service and when needed, the COVID-19 Outpatient Clinic at the National University Hospital in Reykjavik[20]. Those 546 cases represented a range of disease severity from no or mild symptoms, classified in severity category 1 (38.3%), moderate in category 2 (25.6%), more severe symptoms in category 3 (30.6%) and a small group of the most severe cases that were hospitalized in category 4 (5.5%). Associations between T-cell responses and disease severity was tested by linear regression with effects representing differences in the logarithm of T-cell responses in cases per unit of the severity scale. All IFN-$\gamma^+$ (effect range 0.12 to 0.22) and IL-2$^+$ (effect range 0.13 to 0.21) as well as double cytokine-producing TNF-$\alpha^+$ IL-2$^+$ (effect range 0.13 to 0.22) and polyfunctional (effect range 0.11 to 0.22) CD4$^+$ T-cell responses against all four proteins increased with each unit of disease severity (Fig. 3c, d and Supplementary Table 2). Furthermore, the asymptomatic/mild group had higher SARS-CoV-2 reactive CD4$^+$ T-cell responses than controls against all four proteins (effect range 0.48 to 0.80) (Fig. 3f and Supplementary Table 2). Putting this into context with increased T-cell responses observed in cases versus controls, we observed that although CD4$^+$ T-cell responses significantly increased with disease severity of the four groups of cytokine-producing cells mentioned above (effect range 0.11 to 0.22), the increase was considerably smaller than when comparing cases to controls (effect range 0.58 to 1.45) or even when comparing those that got the mildest form of the disease with controls (effect range 0.43 to 1.27) (Supplementary Table 2). Although, SARS-CoV-2 reactive CD8$^+$ T-cell responses did not associate with the severity of acute SARS-CoV-2 infection, IFN-$\gamma^+$ CD8$^+$ T-cells responding to N (effect = 0.89, 95% CI: 0.74, 1.04, $P = 1.1e-28$) and S1 (effect = 0.45, 95% CI: 0.29, 0.61, $P = 3.6e-8$) protein stimulations were increased in those with the mildest form of symptoms compared with controls (Fig. 3e, g and Supplementary Table 3). No significant association was observed between T-cell responses and history of having been diagnosed with any of the major COVID-19 comorbidities following adjustment for multiple testing (Supplementary Data 2).

The SARS-CoV-2 reactive CD4$^+$ and CD8$^+$ T-cell responses were correlated among cases and, to a lesser extent, among controls (Supplementary Fig. 6). When specifically looking at the N protein reactive CD8$^+$ T-cell response in cases (as it was the strongest CD8$^+$ T-cell response), we observed the strongest correlation with the N protein reactive CD4$^+$ T-cell response with Spearman correlation 0.24 (95% CI: 0.16, 0.36) followed by CD4$^+$ T-cells recognizing the S1 0.21 (95% CI: 0.13,0.35), S 0.2

(95% CI:0.11,0.35), and M 0.14 (95% CI: 0.05, 0.37) proteins (Supplementary Fig. 6a).

**SARS-CoV-2 induced T-cell responses were stable at 3 to 8 months after diagnosis.** To examine the longevity of the SARS-CoV-2 induced T-cell responses, we analyzed how the CD4$^+$ and CD8$^+$ T-cell responses changed over time from diagnosis of infection (samples collected up to 259 days from diagnosis) in 90 paired samples collected from the same individuals. The first sample was collected 3–6 months after diagnosis and the second 3–5 months later (Supplementary Fig. 7a). Polyfunctional CD4$^+$ and IFN-$\gamma^+$ CD8$^+$ T-cells were detected up to 8 months from diagnosis with no evidence of their frequency declining within that timeframe for any of the SARS-CoV-2 proteins tested (Fig. 4a, b and Supplementary Fig. 7b, c). The same trend was observed for the double and single cytokine-producing SARS-CoV-2 reactive CD4$^+$ T-cell phenotypes as well as CD8$^+$ T-cells, i.e., no evidence of decline for up to 8 months following diagnosis.

We also collected paired samples from 80 cases and 44 controls, where the first sample was collected before the pandemic and the second sample was collected during the pandemic (Supplementary Fig. 8). For 30 of the 80 cases, we also collected a sample 3–5 months after the first convalescent sample was taken (Supplementary Fig. 9). Interestingly, in cases, only M-reactive polyfunctional CD4$^+$ T-cell responses correlated between samples collected before and during the pandemic with a tendency of individuals with preexisting immunity to have higher responses observed in the samples collected during the pandemic (Supplementary Fig. 8a). In controls, the frequency of polyfunctional CD4$^+$ T-cells responding to the M and S proteins correlated between the paired samples, but there was no indication that the pandemic samples were higher than the pre-pandemic samples (Supplementary Fig. 8c). Although pre-pandemic CD8$^+$ T-cell responses were low, the frequency of SARS-CoV-2 reactive IFN-$\gamma^+$ CD8$^+$ T-cells correlated between samples collected before and during the pandemic, observed both in cases and controls. There was a trend toward N-reactive CD8$^+$ T-cell responses to be higher following SARS-CoV-2 infection in those with measurable N-reactive CD8$^+$ T-cell responses pre-pandemic, but this was not observed for the other proteins or in controls (Supplementary Fig. 8b, d).

**SARS-CoV-2 reactive CD4$^+$ and, to a lesser extent, CD8$^+$ T-cell responses correlated with antibody levels.** Among cases, total antibody levels i.e., IgG, IgM, and IgA (pan-Ig) directed against either the RBD of the S1 subunit (Roche and Wantai) or the N protein (Roche) were highly correlated with each other (the Spearman correlation between the two S1-RBD assays was 0.86 (95% CI: 0.83, 0.89) and between results measured by the two

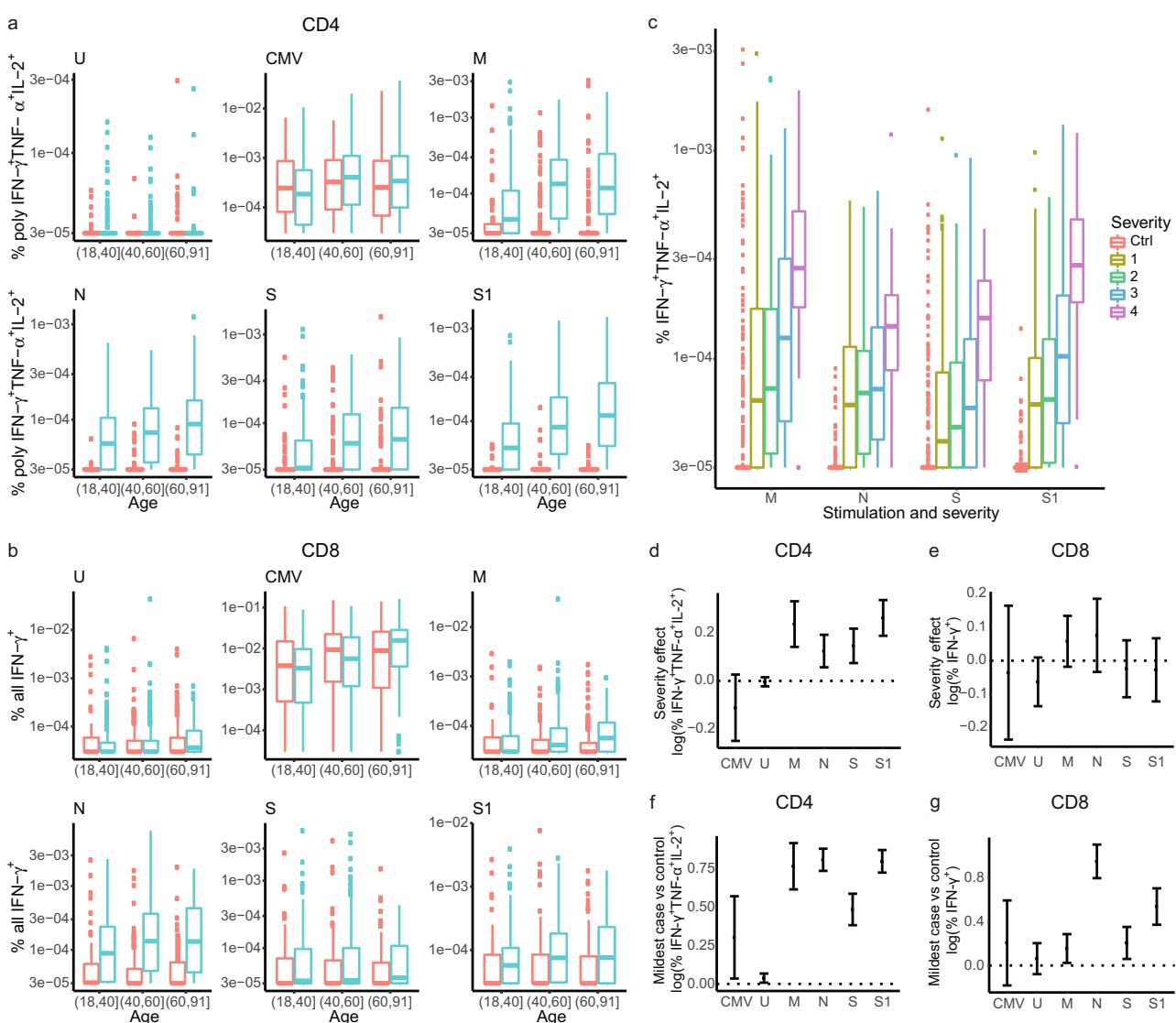

**Fig. 3 SARS-CoV-2 reactive CD4$^+$ T-cell responses correlate with age and disease severity while CD8$^+$ T-cell responses only correlate with age.** Box plots showing the correlation between SARS-CoV-2 reactive **a** polyfunctional CD4+ as well as **b** all IFN-γ$^+$ CD8$^+$ T-cells and age. Frequency of cytokine secreting T-cells (as indicated on the Y-axis) and age (that was split into age bins as indicated on the X-axis) following different stimulations (as indicated above each plot). Uninfected controls are displayed in red (for CD4$^+$: $n$ 18–40 = 64, $n$ 40–60 = 166, and $n$ 60–91 = 160, for CD8$^+$: $n$ 18–40 = 65, $n$ 40–60 = 165, and $n$ 60–91 = 155) and SARS-CoV-2 cases in blue (for CD4$^+$: $n$ 18–40 = 300, $n$ 40–60 = 328, and $n$ 60–91 = 114, for CD8$^+$: $n$ 18–40 = 300, $n$ 40–60 = 327, and $n$ 60–91 = 113). **c** Box plots show the correlation of polyfunctional CD4$^+$ T-cell responses (Y-axis) with disease severity (indicated by the different colors) upon different stimulation conditions (X-axes). For comparison, we show the response induced by each stimulation condition in the uninfected controls (ctrl, $n$ = 392). The frequency of polyfunctional CD4$^+$ T-cell responses in SARS-CoV-2-infected individuals were grouped into four severity groups based on data from questionnaires and/or medical records. 1: Asymptomatic/mild ($n$ = 208); 2: moderate ($n$ sampes = 140); 3: severe ($n$ = 167); 4: hospitalized ($n$ = 30) (**b**). The bottom and top of the boxes correspond to the 25th (Q1) and 75th (Q3) percentiles, the line inside the box corresponds to the median, and the whiskers are located at max(min(Expression), Q1 − 1.5 IQR) and min(max(Expression), Q3 + 1.5 IQR), respectively (where IQR is the interquartile range = Q3 − Q1). **d**, **e** Estimates for differences in the logarithm of the percentage of **d** polyfunctional CD4$^+$ T-cells out of all CD4$^+$ T-cell responding to the different stimulation conditions between unit of the severity scale of the cases and same is shown for **e** all IFN-γ$^+$ CD8$^+$ T-cells for controls ($n$ = 387), 1: Asymptomatic/mild ($n$ = 207); 2: moderate ($n$ sampes = 139); 3: severe ($n$ = 167); 4: hospitalized ($n$ = 30). **f** Difference in the logarithm of the percentage of polyfunctional CD4$^+$ T-cells out of all CD4$^+$ T-cell responding to the different stimulation conditions between cases with the mildest disease severity ($n$ = 208) and uninfected controls ($n$ = 392). **g** Difference in the logarithm of the percentage of polyfunctional CD8$^+$ T-cells out of all CD8$^+$ T-cell responding to the different stimulation conditions between cases with the mildest disease severity ($n$ = 207) and uninfected controls ($n$ = 387). The error bars indicate 95% confidence intervals (CI). See also Tables S2 and S3. Association between T-cell responses in different groups was tested by linear regression and significance was assessed with the standard linear regression $t$-test.

S1-RBD assays and the N protein assay were 0.58 (95% CI: 0.52, 0.63) and 0.67 (95% CI:0.63, 0.72)). The frequency of SARS-CoV-2 reactive CD4$^+$ T-cells responding to all four proteins correlated with all three pan-Ig assays (the Spearman correlation ranged from 0.18 to 0.37 for polyfunctional CD4$^+$ T-cells; Supplementary Fig. 10a). SARS-CoV-2 protein CD8$^+$ T-cell responses did, however, not correlate with the pan-Ig levels directed against either N or S protein (Supplementary Fig. 10b).

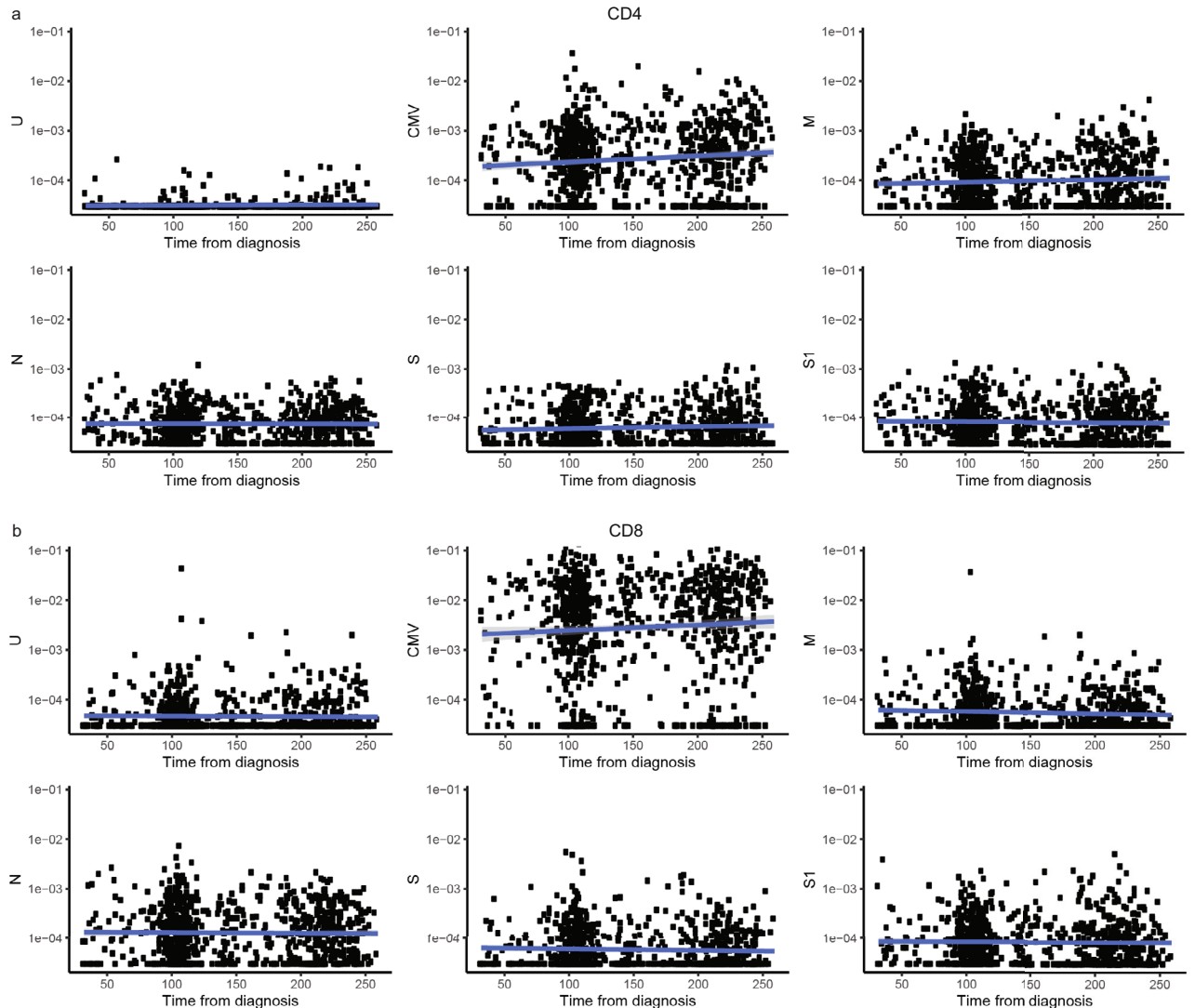

**Fig. 4 SARS-CoV-2 induced T-cell responses do not decline in the first 8 months after diagnosis.** Cross-sectional analysis of SARS-CoV-2 reactive **a** polyfunctional CD4$^+$ and **b** all IFN-γ$^+$ CD8$^+$ T-cells from all subjects in the study where samples were collected up to 259 days from diagnosis. Y-axes shows percentage of polyfunctional **a** CD4$^+$ ($n = 862$ samples) and **b** CD8$^+$ ($n = 859$ samples) T-cells responding to the stimulation indicated on the Y-axes (U, CMV, M, N, S, or S1). Y-axes indicate time from diagnosis in days. The solid blue line indicates the logistic-regression line and the gray area around the blue line indicates the 95% confidence intervals (CI). Correlation between T-cell responses was estimated using Spearman's correlation coefficient and a Jackknife method was used to calculate confidence intervals.

In order to further test the functionality of the antibodies, we measured ACE2 blocking capacity in a subset of our samples ($n = 538$) using an MSD-ECLIA. This assay measures the ability of antibodies to inhibit the binding of labeled soluble ACE2 to plate-bound Spike protein and has been shown to correlate with live and pseudovirus neutralization assays[21]. The total antibody levels and the ACE2 blocking effect correlated best when measuring the pan-Ig against S1-RBD (Roche assay: 0.77 (95% CI: 0.73, 0.82), Wantai assay: 0.70 (95% CI: 0.64, 0.76) followed by pan-Ig against the N protein (Roche; 0.60 95% CI: 0.54, 0.66; Supplementary Fig. 11). We observed a significant correlation between the frequency of polyfunctional SARS-CoV-2 reactive CD4$^+$ T-cells responding to all four proteins and ACE2 inhibitory antibodies (the Spearman correlation ranged from 0.26 to 0.46) with the strongest correlation observed between S1-reactive polyfunctional CD4$^+$ T-cells and level of ACE2 inhibitory antibodies (Fig. 5a). In contrast to the lack of correlation between CD8$^+$ T-cell responses and pan-Ig antibody levels we detected a significant correlation between SARS-CoV-2

reactive CD8$^+$ T-cell responses and ACE2 inhibitory antibodies (the Spearman correlation ranged from 0.09 to 0.17), although the correlation was not as strong as observed for the CD4$^+$ T-cell responses. For CD8$^+$ T-cells the S1 and N-reactive responses had the strongest correlation with ACE2 inhibitory antibodies (Fig. 5b).

The polyfunctional CD4$^+$ T-cell responses correlated well with the positivity of both the qPCR and pan-Ig antibody assays (Supplementary Fig. 12). Of individuals with two or more positive pan-Ig results, 79% had a marked (above 0.003%) polyfunctional CD4$^+$ T-cell response, compared to 6% of those with one or no positive pan-Ig results ($P < 0.0001$).

## Discussion

Here we report analyses of SARS-CoV-2 reactive T-cell responses from a large number of well-defined cases ($n = 768$), with a wide range of disease severity, compared to uninfected controls ($n = 500$), recruited both before and longitudinally during the

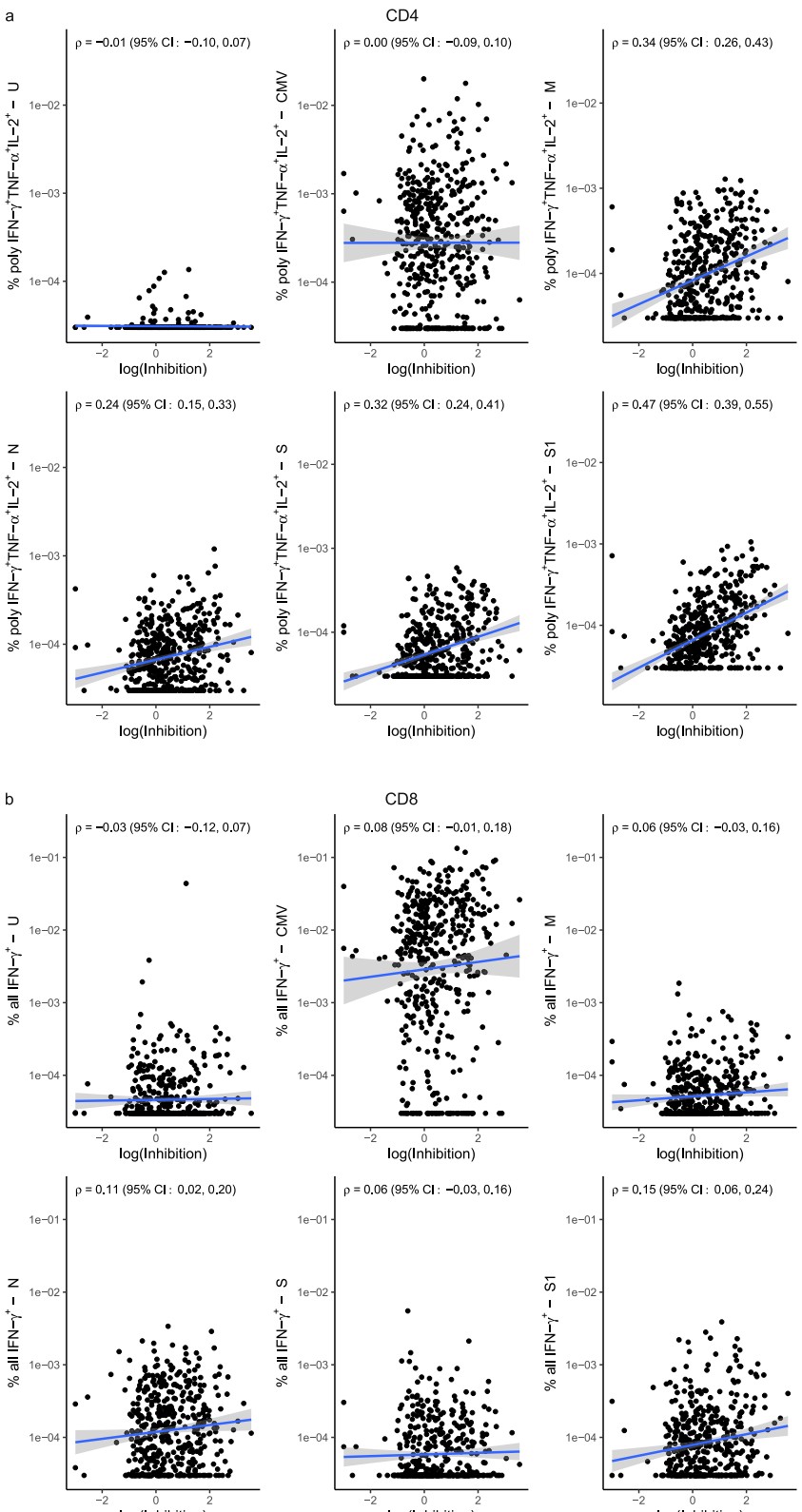

**Fig. 5 SARS-CoV-2 reactive CD4$^+$ and, to a lesser extent, CD8$^+$ T-cell responses correlate with antibody levels.** Scatter plot show a correlation between **a** polyfunctional CD4$^+$ as well as **b** all IFN-$\gamma^+$ CD8$^+$ T-cells and functional antibody responses. Inhibition of the antibodies was measured using an MSD-ECLIA-based ACE2 blocking assay and reflects how well the plasma antibodies inhibit binding between soluble ACE2 and plate-bound S1-RBD. Inhibition efficacy of the antibodies is given on a log scale on the X-axes and percentage **a** polyfunctional CD4$^+$ T-cells out of all CD4$^+$ T-cells ($n = 456$ convalescent SARS-CoV-2 cases) and **b** all IFN-$\gamma^+$ CD8$^+$ T-cells out of all CD8$^+$ T-cells are given on the Y-axes ($n = 454$ convalescent SARS-CoV-2 cases). The solid blue line indicates the logistic-regression line and the gray area around the blue line indicates the 95% confidence intervals (CI). The correlation between T-cell responses and functional antibody responses shown in **a**, **b** was estimated using Spearman's correlation coefficient and a Jackknife method was used to calculate confidence intervals.

pandemic, up to eight months after its onset in Iceland. We observed robust CD4$^+$ T-cell responses following SARS-CoV-2 infection upon stimulation with selected domains of the S protein (see methods for details), S1 subunit, M and N proteins in line with previous reports[6,9–13,22,23]. The CD8$^+$ T-cell responses to all four SARS-CoV-2 proteins were also significantly higher in the cases than in controls, although more modest than CD4$^+$ T-cell responses, consistent with other studies[9,11,12,23]. The strongest CD8$^+$ T-cell responses were observed against the N protein, making it promising as a vaccine component to induce cytotoxic T-cell responses. Our analysis revealed relatively stable SARS-CoV-2 reactive CD4$^+$ and CD8$^+$ T-cell responses during the timeframe of our study (i.e., three to eight months from SARS-CoV-2 infection), with no indication of decline during this period, in agreement with the previous reports[9,24].

To our knowledge, this is the most comprehensive study on the association between different HLA alleles and SARS-CoV-2 reactive T-cell responses following infection. Our results clearly demonstrated a highly significant association of CD8$^+$ T-cell responses with several HLA alleles, most significant between increased N-reactive CD8$^+$ T-cell responses and HLA-B*07:02. We observed that many of the N-derived peptides predicted to bind to HLA-B*07:02 are well conserved between SARS-CoV-2 and other members of the *Coronaviridae*. In line with our results, several recent papers have reported immunodominant N-derived peptides when presented by the HLA-B*07:02 allele[25–28] and cytotoxic T-cell response directed against N $_{105-113}$-B*07:02 was associated with less severe COVID-19 disease. Interestingly, a high frequency of naïve CD8$^+$ T-cell precursors recognizing N$_{105-113}$-HLA-B*07:02 was also found in pre-pandemic samples[28,29]. It is therefore tempting to speculate that the strong N-reactive CD8$^+$ T-cell responses observed in carriers of HLA-B*07:02 in our nationwide study population reflect an expansion of preexisting CD8$^+$ T-cell immunity. Preexisting immunity could affect COVID-19 susceptibility and/or severity, as implied in a recent paper suggesting that preexisting T-cell immunity may prevent highly exposed healthcare workers from contracting infection[30]. Although we and others[31] found less convincing evidence of SARS-CoV-2 reactive CD8$^+$ than CD4$^+$ T-cell responses in the blood of uninfected individuals, a recent report indicates that tissue-resident SARS-CoV-2 cross-reactive CD8$^+$ T-cell immunity might be found in the respiratory tract of uninfected individuals although not found in circulation[32]. Further, we cannot exclude that the experimental setup used in our study, namely short in vitro activation with 15-mer peptides, might underestimate the circulating CD8$^+$ T-cell responses. Preexisting T-cell responses have also been reported to target non-structural proteins[11,30], not tested here. HLA-A*01:01 was the only allele found to associate with lower CD8$^+$ T-cell responses. Interestingly, at least two studies have reported the association between HLA-A*01:01 and increased disease severity[33,34] and our results indicate that this might, at least partially, be explained by the fact that carriers of this HLA allele form lower S1-reactive CD8$^+$ T-cell responses.

The CD4$^+$ T-cell responses to all four SARS-CoV-2 proteins correlated positively with age, disease severity, and humoral responses directed against SARS-CoV-2. We cannot exclude that the association of increased CD4$^+$ T-cell responses with increased severity and age is due to polyclonal and low avidity T-cells, as reported by ref. [35] although we observed a significant increase in double and triple cytokine-producing cells, indicating high functionality of those cells. Further, the increase in the frequency of antigen-specific T-cells in higher disease score patients was not explained by prolonged lymphopenia as was described in the acute phase of COVID-19[36] as we observed similar results when testing for association with absolute counts of antigen-reactive

T-cells (Supplementary Table 2). Therefore, the association between increased CD4$^+$ T-cell responses in convalescent individuals and disease severity likely reflects more abundant and prolonged antigen stimulation and consequent expansion of the adaptive immune response during the acute phase of the infection. CD8$^+$ T-cell responses were only correlated with age. However, when testing the functionality of the antibodies, we observed a significant correlation between CD8$^+$ T-cell responses against the S1 subunit and the N protein and the ability of the antibodies to inhibit the binding between the ACE2 receptor and Spike protein. Therefore, our data indicate that individuals with highly functional S1-RBD specific antibodies also have cell-mediated immunity that is mainly directed against the N and S1 proteins of the virus.

Polyfunctional CD4$^+$ T-cells are known to be functionally superior to single cytokine-producing cells and correlate with protection against intracellular pathogens such as *Mycobacterium tuberculosis* and the influenza virus[37–40]. Importantly, we detected polyfunctional SARS-CoV-2 reactive CD4$^+$ T-cells against all four proteins as well as N and S1-reactive IFN-γ CD8$^+$ T-cells in the least severe cases, indicating that even very mild disease induces some level of protective T-cell responses, although the level of T-cell responses required for protection against COVID-19 is unknown.

SARS-CoV-2 reactive memory T-cell responses have been reported months after SARS-CoV-2 infection, in the absence of detectable circulating antibodies, raising the possibility that antibody levels may underestimate the extent of adaptive immune responses[12]. We found no evidence of individuals with a history of positive qPCR test results having SARS-CoV-2 specific T-cell responses despite remaining seronegative within the first eight months after infection. Seronegative, qPCR-positive individuals did not have a T-cell response distinguishable from that of uninfected individuals, suggesting that they were false qPCR-positive and that there is an overall good concordance between the induction of humoral and T-cell responses to SARS-CoV-2 following infection.

Taken together, in a large nationwide study, we examined the heterogeneity of T-cell responses following infection with SARS-CoV-2 and demonstrated that they correlated with HLA type, disease severity, age and humoral responses. The fact that S1-reactive CD4$^+$ T-cells showed the strongest correlation with functional antibodies underscores the ability of the current Spike-based vaccines to activate both CD4$^+$ T- and B-cells in line with their induction of protective humoral immunity. However, given the emergence of SARS-CoV-2 variants exhibiting reduced sensitivity to vaccine-elicited antibodies, our results provide valuable insight into the HLA restriction of CD8$^+$ T-cell mediated immunity following natural infection that could help to guide the development of the next generation of SARS-CoV-2 vaccines.

## Methods

**Ethical considerations**. Written informed consent was obtained from all participants in accordance with the Declaration of Helsinki, and the study was approved by the National Bioethics Committee (VSN_20-076) after review by the Icelandic Data Protection Authority (DPA). Personal identifiers were encrypted by a third-party system approved and overseen by the DPA[41] and all processing of personal data were in agreement with conditions set by the DPA (PV_2017060950ÞS).

**Study design and participants**. We collected blood samples from 768 convalescent SARS-CoV-2 cases and 500 uninfected controls. For 148, we had collected samples before the pandemic (October 2001–February 2020). Eighty of the pre-pandemic samples were from SARS-CoV-2 cases from whom we also had samples following the infection (May–December 2020) and 44 were from controls from whom we also had samples collected during the pandemic (June–December 2020; Table 1). For 90 of the cases, we collected paired samples during the pandemic.

**Determination of SARS-CoV-2 status**. We measured antibody levels using three pan-Ig antibody tests and required at least two out of three to be positive for an individual to be considered a SARS-CoV-2 case We required control samples collected in 2020 to be negative for all three antibody assays (Table 1).

**qPCR measurements**. RNA from nasal swabs were extracted and tested for the presence of SARS-CoV-2 infection using quantitative real-time PCR (qRT-PCR) methods. RNA extraction and qRT-PCR were performed either at the Department of Clinical Microbiology laboratory at Landspitali—the National University Hospital of Iceland (LUH) or at deCODE with similar methods[42]. Extraction methods include cell lysis and Proteinase K treatment using an automated magnetic bead-purification procedure. Chemagic Viral RNA kit was used to extract viral RNA at deCODE using the Chemagic360 instrument (Perkin Elmer), whereas MagNA Pure LC 2.0 or MagNA Pure Compact instruments (Roche LifeScience) were used for extraction at LUH.

LUH used a single probe pan screening assay for a conserved region of the E-gene of betacoronaviruses, followed by confirmatory measurements for all positive samples using a nCoV-2019 specific assay based on the RdRP gene or a three probe based TaqMan™ Fast Virus 1- step Master Mix, 2019-nCoV Assay kits v1 (Thermo Fisher Scientific). All labeled probes and primers for the E and RdRP genes were from TAG (Copenhagen, Denmark). 2019 E-gene control and SARS-CoV Frankfurt 1 positive controls were obtained from EVAg (https://www.european-virus-archive.com/bundle/diagnostics-controls-wuhan-coronavirus-2019-2019-ncov). Each assay was done in a 25 μL total sample volume with FAM™ dye-labelled probes in addition to VIC™ dye labeled probes for human RNaseP as an internal control. 96 well plates were scanned in an AB-7500 Fast real-time PCR thermocycler for 40 cycles of amplification (Thermo Fisher Scientific). Ct values <35 were considered strong positive in the E-gene screening assay and went for confirmatory testing using RdRp, whereas samples with Ct values between 35–37 were confirmed using the TaqMan™ Fast Virus method. Samples with Ct values from 37-40 were classified as inconclusive and were tested again to confirm their status.

deCODE used exclusively the three probe TaqMan™ Fast Virus 1-step Master Mix, 2019-nCoV v1 assay. Hamilton STARlet 8-channel liquid handler was used at deCODE for aliquoting and mixing the samples and the plates were scanned in an ABI 7900 HT RT-PCR system with a total of 40 cycles of amplification. Ct values <37 in at least two of three assays included in the Fast Virus 1-step Master Mix, 2019-nCoV v1 assay, were classified as positive. Ct values between 37 and 40 were classified as inconclusive and testing of those samples was repeated. Repeated testing giving the same result with at least two probes resulted in a positive classification of the sample. If repeated testing gave positive results for only one probe, the test was considered inconclusive, and a new sample from the subject was requested. Samples with undetected FAM™ dye Ct values or values equal to 40 in all three assays were classified as negative if the human RNaseP assay was positive (VIC™ dye Ct <40).

**Serological measurements**. Serum samples were measured using pan-immunoglobulin (pan-Ig: IgM, IgG, and IgA) assays against the N protein (Roche ECLIA) and the RBD of the S1 subunit of the S protein (Wantai ELISA and Roche ECLIA) according to protocols supplied by the assay manufacturers[2] (Table 3).

A multiplexed MSD immunoassay (Meso Scale Diagnostics, LLC #K15436U) was used to measure the inhibition of ACE2 binding to the Spike protein by human serum samples. The MSD immunoassays were performed according to the manufacturer's protocol. To measure ACE2 inhibition, pre-coated plates were blocked with MSD Blocker A for 30 min. Following a wash with MSD wash buffer, serum samples (diluted 1:10 and 1:100 in diluent buffer), were added to the plates. After 1 h incubation, Sulfo-tag labeled ACE2 was added to the wells and allowed to incubate for 1 h. Plates were washed, MSD GOLDTM Read Buffer B added, and plates immediately read using a MESO® SECTOR S 600 Reader.

The assay includes a SARS-CoV-2 Spike monoclonal neutralizing antibody, which is used to generate a calibration curve for each plate. The calibration curve was to calculate neutralizing antibody concentrations in samples (in units/mL) by backfitting the measured signals to the curve.

**Disease severity**. All Icelanders diagnosed with SARS-CoV-2 infection by qPCR were monitored by the telehealth monitoring service (TMS) of the LUH COVID-19 Outpatient Clinic in Reykjavik, Iceland[20]. We grouped the cases into four categories according to the severity of the acute infection, severity classification by the TMS, level of care at LUH and self-assessment of symptoms. Out of the 768 SARS-CoV-2 cases in this study, we had information on disease severity for 546 (71.0%). Those 546 cases represent a range of disease severity; 209 (38.3%) grouped into the least severe group (severity category 1), 140 (25.6%) in category 2, 167 (30.26in category 3 and a small group of the most severe cases 30 (5.6%) in category 4.

**Sample collection and treatment**. PBMCs were isolated from venous blood samples via standard Ficoll-Paque density gradient centrifugation at 800 G for 15 min in 50 ml Blood-Sep spin tubes and cryopreserved in liquid nitrogen. Prior to use, cells were thawed and incubated overnight at 37 °C and 5% $CO_2$ at $1.5 \times 10^7$ cells/mL in RPMI 1640 supplemented with 10% fetal bovine serum (FBS) and 1x

Penicillin-Streptomycin. After resting overnight, cells were filtered, counted and split into $2 \times 10^5$ cells per well to be stimulated with 0.5 μg/mL SARS-CoV-2 Prot_M, _N, _S, _S1 or CMV peptides pools along with 0.3 μg/mL CD28 and CD49d antibodies co-stimulation for 8 h total, 5 μg/mL Brefeldin A and Monensin were added at 5 μg/mL after 30 min (Table 3). We note that only selected sequence domains (aa 304–338, 421–475, 492–519, 683–707, 741–770, 785–802, and 885–1273) are included in the S peptide stimulation pool.

**HLA genotyping**. HLA alleles were genotyped using GraphTyper, a publicly available algorithm and software[43] in a set of 63,460 WGS individuals. Sequence reads were aligned to sequences in the HLA alleles found in the IPD-IMGT/HLA database[44]. The genotypes were then imputed into a set of 173,025 microarray typed individuals[45].

**Prediction of peptide-MHC class I binding**. NetMHCpan v4.0 server[18] was used to predict the binding of SARS-CoV-2 derived peptides to any of the significantly associating MHC alleles. The method is trained on a combination of more than 180,000 quantitative binding data and MS-derived MHC eluted ligands. The binding affinity data covers 172 MHC molecules from humans (HLA-A, B, C, E). The SARS-CoV-2 and CMV protein sequences were obtained from NCBI and used to generate all possible peptides of 8–14-mer lengths. N (QHD43423.2) derived peptides were used to derive binding scores to HLA-B*07:02, S1 derived peptides (containing amino acid 1–692 of the S protein; QHD43416.1) were used to derive binding scores to HLA-C*07:02 and HLA-A*01:01 and CMV derived peptides were used to derive binding scores to HLA-C*07:02.

**Multiple sequences alignment**. ConSurf server (https://consurf.tau.ac.il/) was used to determine conservation patterns and scores of the S and N protein of SARS-CoV-2[46]. Briefly, a multiple sequences alignment of 150 homologous sequences was constructed using MAFFT. The Bayesian algorithm was used to compute position-specific conservation scores that were divided into a discrete scale of nine grades. The conservation scores were projected onto the SARS-CoV-2 Spike protein in the closed state (PDB ID 6VXX) as a reference.

**Flow cytometry**. Cells were washed in phosphate-buffered saline (PBS), Fc receptors blocked with TruStain FcX and viable cells identified by exclusion using LIVE/DEAD Fixable Aqua Dead Cell Stain Kit (Table 3). Cells were washed again with PBS supplemented with 2% FBS (FACS buffer) and surface markers were detected via the addition of directly conjugated CD3-APC-Cy7, CD4-BV605, and CD8-PE-Cy7 antibodies at pre-titrated concentrations for 20 min at room temperature. Cells were then washed again in FACS buffer and fixed/permeabilized according to the manufacturer's instructions using a FoxP3/Transcription Factor Staining Buffer Set. Intracellular markers were detected via the addition of directly conjugated IFN-γ-FITC, TNF-α-APC and IL-2-PE antibodies at pre-titrated concentrations in permeabilization buffer for 20 min at room temperature. Samples were acquired using an Attune NxT. Data were analysed with FlowJo software version 10.7.1. Core gates included singlet isolation (FSC-H versus FSC-A), live CD3 selection (CD3 versus Aqua), lymphocyte enrichment (SSC-A versus FSC-A), and CD4 or CD8 selection (CD4 versus CD8).

**Automatic gating**. Flow cytometry measurements were processed by the automatic gating algorithm. The R script (available at https://www.decode.com/summarydata) was designed to emulate the standard practices used when manually gating and was customized to fit the properties of this project. To allow practitioners to visually verify the results and decisions made by the algorithm, the script generates gating images for each cytometry experiment. Antigen-reactive production of IFN-γ-FITC (IFN), TNF-α-APC (TNF) and IL-2-PE (IL-2) for all cells (events) classified as CD4 or CD8 were written as intermediate results to enable fine-tuning of cutoffs for positive cytokine response.

The events measured are gated (filtered) by a series of gates in the following order (as illustrated in Supplementary Fig. 13).

Singlet gate: To detect singlets from double cells, we applied linear regression of the area of forward scatter (FSC_A) to predict the height of forward scatter (FSC_H) with the intercept fixed at zero. We assumed that the residual (r) of the linear prediction was normally distributed and classified events as doublet if $|r| > Z_{0.01}$. This method was repeated using the events classified in the first pass as singlet and the regression model was modified with no constraints on the intercept. Again, events were classified as doublets when the absolute residual was higher than $Z_{0.01}$.

Scatter gate: To detect events corresponding to lymphocytes, we use two channels: the area of the forward (FSC_A) and the area of the side scatter (SSC_A). We applied Gaussian Mixture Models (GMM)[47] on log-transformed channel levels using either two or three clusters setup. Overlap of clusters and closeness of center was used to determine the best GMM setup. The GMM cluster closest to (200, 200) were classified as lymphocytes. To evaluate class boundaries, we used means and variance from GMM to calculate Mahalanobis distance (MD)[48]. Assuming MD are chi-square distributed, the initial lymphocytes class boundaries were set to $\chi_2^2$ $p = 0.1$. If there was a high overlap of events classified by GMM to the cluster closer to (0,0) and the events classified as lymphocytes with MD, we shrunk the MD cutoff down iteratively with $p = 0.05$ steps until less than 10% of events would overlap. If GMM cluster centers were too close or not in increasing order in both

**Table 3 Key resources table.**

| Reagent or resource | Source | Identifier |
|---|---|---|
| **Chemicals and Peptides** | | |
| Ficoll-Paque Plus | GE Healthcare | #17144002 |
| RPMI 1640 Medium, GlutaMAX Supplement | Thermo Fisher Scientific | #61870044 |
| Fetal Bovine Serum | Thermo Fisher Scientific | #10500064 |
| Penicillin-Streptomycin (10,000 U/mL) | Thermo Fisher Scientific | #15140148 |
| DPBS, no calcium, no magnesium | Thermo Fisher Scientific | #14190169 |
| Brefeldin A Solution (1,000X) | Biolegend | #420601 |
| Monensin sodium salt | Biotechne | #5223 |
| PepTivator® SARS-CoV-2 Prot_M | Miltenyi | #130-126-703 |
| PepTivator® SARS-CoV-2 Prot_N | Miltenyi | #130-126-699 |
| PepTivator® SARS-CoV-2 Prot_S | Miltenyi | #130-126-701 |
| PepTivator® SARS-CoV-2 Prot_S1 | Miltenyi | #130-127-048 |
| PepTivator® CMV pp65, human | Miltenyi | #130-093-435 |
| **Antibodies** | | |
| Ultra-LEAF™ Purified anti-human CD28 Antibody (1 µg/ml) | Biolegend | #302934 |
| Ultra-LEAF™ Purified anti-human CD49d Antibody (1 µg/ml) | Biolegend | #304340 |
| Human TruStain FcX™ | Biolegend | #422302 |
| APC/Cyanine7 anti-human CD3 Antibody (1:200) | Biolegend | #300318 |
| Brilliant Violet 605™ anti-human CD4 Antibody (1:200) | Biolegend | #300556 |
| PE/Cyanine7 anti-human CD8a Antibody (1:200) | Biolegend | #301012 |
| FITC anti-human IFN-γ Antibody (1:100) | Biolegend | #502506 |
| APC anti-human TNF-α Antibody (1:100) | Biolegend | #502912 |
| PE anti-human IL-2 Antibody (1:100) | Biolegend | #500307 |
| **Biological samples** | | |
| Peripheral blood mononuclear cells from indicated individuals | This manuscript | N/A |
| Serum from indicated individuals | This manuscript | N/A |
| Nasopharygeal swab samples from indicated individuals | This manuscript | N/A |
| **Critical commercial assays** | | |
| LIVE/DEAD Fixable Aqua Dead Cell Stain Kit | Thermo Fisher Scientific | #L34957 |
| eBioscience™ Foxp3 / Transcription Factor Staining Buffer Set | Thermo Fisher Scientific | #00552300 |
| SARS-CoV-2 Ab Elisa | Nordic Biosite | #256-WS-1096-96 |
| Elecsys Anti-SARS-CoV-2 S | Roche | #09 289 267 190 |
| Elecsys Anti-SARS-CoV-2 N | Roche | #09 203 095 190 |
| TaqPath™ COVID-19 CE-IVD RT-PCR Kit | Thermo Fisher Scientific | # A48067 |
| V-PLEX SARS-CoV-2 Panel 6 (ACE2) kit | Meso Scale Diagnostics | #K15436U |
| **Software** | | |
| Attune Nxt | Thermo Fisher Scientific | v.3.2.1 |
| FlowJo | BD | v.10.7.1 |

channels, the algorithm reverted to one class solution where all events below (80,80) were removed and MA calculated for the rest of the events. Finally, the lymphocyte class boundary was defined as $\chi_2^2$ $p = 0.3$.

CD3 gate: To detect events corresponding to live CD3 cells, we used the Live-Dead fixable aqua (LD) and CD3 channel. Similarly to the scatter gate, we used two class GMM and MD, with a cutoff at $\chi_2^2$ $p = 0.0001$. In addition, we added a hard cutoff for minimum CD3 values ($355 < CD3$) and a cutoff for max LD, defined relative to cluster center and variance.

CD4 vs CD8 gate: To distinguish between CD4 and CD8 cells, we start by excluding high and low extremes of CD4 and CD8 channels. Then we used two class GMM to evaluate the means and variance of both clusters. Again, we used MD to assign the probability of events belonging to the CD4 or CD8 class. If there was high overlap in the two clusters, we scaled the probability of the class which had the higher variance. Using the class probability estimates and the mixing proportion which GMM provides, we assigned each event to be CD4, CD8, or ambiguous. Finally, we defined hard cutoffs (see code) to assign events as ambiguous in the bottom-left and upper-right area of the gating plot. To eliminate ambiguous events, we found the line passing through the origin that separates CD4 and CD8 clusters. Events that were originally labeled as CD4, but crossed the diagonal line towards CD8 are labeled ambiguous, and vice versa.

Cutoff for cytokine response: There was considerable variability in median cytokine level between measurements (flow cytometry experiments). Also, high variability of the deviations (log median absolute deviation) in cytokine levels within cytometry experiments when compared between cytometry experiments. Based on the mean $CD4^+$ and $CD8^+$ T-cell counts of all participants with the 6 stimulation types tested, we defined a QC threshold of 20,000 cells and 10,000 cells as a cutoff for $CD4^+$ T and $CD8^+$ T-cell counts, respectively. Based on this threshold, 4 and 12 samples were excluded from $CD4^+$ and $CD8^+$ T-cell analysis, respectively.

**Statistics and reproducibility**. We used a likelihood ratio method to calculate confidence intervals of fractions with the Clopper-Pearson exact method when the estimated fraction was 0 or 1 (https://CRAN.R-project.org/package=binom). We estimated the correlation between T-cell responses and other variables using Spearman's correlation coefficient and used a Jackknife method to calculate confidence intervals[49]. When testing for association with T-cell responses, we performed multiple linear regression, using the logarithm of T-cell responses as a response and using age and sex as covariates—except when testing for association with these variables. Sex was included as an indicator variable (0 for males and 1 for females). Significance was assessed with the standard linear regression $t$-test with $n - p$ degrees of freedom, where $n$ is the number of available observations and $p$ is the number of parameters (2 or 3). The adjusted $R^2$ for each linear model is shown in the supplementary tables.

Since T-cell responses are highly skewed and their variance depends on the T-cell response itself, we performed additional analysis using general least squares models (as implemented in the gls function in R), where we allowed the variance to depend linearly on each of the covariates included in the model using the varFixed() option. Overall, the results did not differ substantially between the simple linear regression and general least squares analysis.

We also tested for association with the absolute number of responding T-cells, without dividing by the number of $CD4^+$ or $CD8^+$ cells, using the quasi-Poisson model implemented in the glm R function. As for the linear regression test, we used age and sex as covariates.

In the genetic association analysis, where we associated T-cell responses with HLA genotypes, we used inverse normal standardization to convert the T-cell responses to a standard normal distribution and then performed standard genetic association testing using logistic regression[50]. To convert effect estimates from standard deviations to the logarithm of T-cell responses, we multiplied the effect

estimates in standard deviations by the standard deviations of the logarithm of T-cell responses.

**Reporting summary**. Further information on research design is available in the Nature Research Reporting Summary linked to this article.

## Data availability

The HLA—T-cell responses summary statistics and R script designed for automatic gating of flow cytometry data are available at [https://www.decode.com/summarydata]. The authors declare that the data supporting the findings of this study are available within the article, its Supplementary Information file, and upon reasonable request.

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

## Acknowledgements
We thank all of the participants that contributed samples for this study for their invaluable contribution to the research. We also thank our research staff at the Patient Recruitment Center for their thorough work.

## Author contributions

T.A.O., K.B., G.L.N, U.T., I.J. D.F.G., and K.S. conceived and designed the study and take responsibility for the integrity of the data and the accuracy of the data analysis. J.S. contributed to the collection of samples and sample handling. K.B., K.G., A.O.A., and F.T. performed cellular stimulations and flow cytometry experiments. G.H.H., P.M., K.B., G.L.N., and D.F.G. prepared and analyzed the flow cytometry data with manual and autogating methods. E.I., T.O., G.M., G.T., H.H., P.S., and S.S. contributed to data preparation and handling. H.P.E. and B.V.H. performed HLA genotyping. D.F.G. performed the statistical analysis. M.K., E.E., D.H., H.L.R., M.I.S., R.F.I., S.B., I.O., and R.P. collected phenotype information on the acute infection. T.A.O., K.B., and D.F.G. drafted the manuscript. All authors had full access to all of the data in the study. All authors critically revised the manuscript for important intellectual content and accept responsibility for the version submitted for publication.

## Competing interests

A.O.A., B.V.H., D.F.G., E.I., T.O., F.T., G.H.H., G.L.N., G.M., G.T., H.H., H.P.E., I.J., J.S., K.B., K.G., P.M., P.S., S.S., T.A.O., U.T., and K.S. are employees of deCODE genetics/Amgen Inc. The remaining authors declare no competing interests.
