## [Peer Review File · Communications Biology]

Reviewers' comments:

Reviewer #1 (Remarks to the Author):

In their manuscript, Olafsdottir et al. analyze CD4 and CD8 T cell responses in healthy controls and SARS-CoV-2 infected individuals. Frequencies of antigen-specific T cells are determined based on cytokine production after stimulation with 4 CoV-2 proteins (M, N, S, S1 domain of S) and CMV pp65 as control. Results are correlated to age, sex and COVID-19 disease severity. Furthermore, based on HLA haplotype analysis, associations between CD8 responses and MHC-I gene expression are studied. The authors end with longitudinal analyses, addressing how T cell immunity is maintained or altered months after infection.

Clearly, the main selling point of the manuscript is the number of individuals included, exceeding by far most, but not all studies focusing on FACS based T cell analysis. Also the correlation of HLA haplotypes and CD8 responses has not been conducted in such depth. SARS-CoV2 related studies have fueled interest in large cohorts that are studied in great detail; in that respect, Bacher et al. extensively analyzed 120 individuals (Bacher, P, Immunity, 2020), while Loyal et al. included more than 500 individuals for CoV2-specific T cell analysis, including MHC-II haplotype analysis (Loyal, L, Science, 2021).

There are several main points of concern regarding the quality of the manuscript:

1) The main focus of the study is the analysis of multi- or polyfunctional T cells. The authors state that on average, 68k CD4+ T cells and 32k CD8+ T cells were acquired. Looking e.g. at Fig. 1C, mean frequencies of polyfunctional CD4+ T cells are usually below 1 in 10.000 – meaning that on average, less than 6.8 cells fall in this category. Based on this, it is doubtful if the assay design regarding starting cell numbers will adequately allow to quantify these very rare cells and ensuing conclusions and correlation analyses in Fig. 2. This problem becomes even more obvious for CD8+ T cells, where many responses are between 1 in 10.000 and 1 in 100.000. Since on average, only 32k cells were analyzed, this results in 3.2 to 0.32 specific cells. The methods part does not state how many cells were used per stimulation; if the mentioned 1×10^7 were used per stimulation condition, recovery after thawing would be extremely poor.

2) What is still incompletely understood is the relation of immune parameters and disease severity. This aspect, together with the influence of age, is addressed in Fig. 4. Since leuko- or lymphopenia has been identified as a hallmark of severe COVID, it would be helpful not only to address frequencies of antigen-specific T cells, but also absolute counts, providing information on whether higher disease score patients harbor also higher absolute antigen specific T cell numbers. Also data is missing on how a positive correlation between CD4+ T cell numbers and severity is addressed; the text p12, l219 states a positive correlation e.g. in Fig 4C, but no test is mentioned or illustrated here.

3) Overall, it is not easy to follow statistical analyses as presented in the paper and judge if the reported results are meaningful. For instance, effect ranges or, alternatively, effects are calculated without extensively naming and explaining the methods for an unexperienced reader. Supporting information is missing allowing basic estimates for weak vs strong effects or ranges. Inter alia, the reported effects (for example starting from line164) and effect ranges (for example from line147) need to be explained more clearly and names of the test system included. It is also not clear how the adjusted analysis (line687) is performed - a linear model is described but 'sex' as a non-parametric variable listed - was a glm used? There are no R^2 /adj R^2 and DFs mentioned. In addition, it is not reported that the requirements of the used methods are fulfilled (such as homoscedasticity for the multivariate model). Overall, the statistical methods part needs to be improved and elaborated. Regarding the figures, they should be reduced in number and be summarized. For an example, the boxplots on P. 48 - 49 could be merged together. In addition, the presentation in age ribbons appears not meaningful since a metric scale would allow a better illustration of data.

4) Given that T cell responses have been shown to depend on patient predisposition, it would be

helpful to correlate responses with comorbidities and not only with age; this information will likely be available for the cohort and greatly enrich pioneering studies as the one from Sattler et al. (Sattler, JCI, 2020) that might also serve as a basis for discussion of this aspect.

In summary, it appears that the authors also see their study as a reproduction of existing studies with a larger cohort. This is reflected by that fact that each of their summary sentences in the 1st paragraph of the discussion (p16) ends with references stating similar findings. This applies to T cell targets, CD4/CD8 response characteristics, longitudinal follow up and correlations of T cell with antibody responses. One of the most important questions, namely, how T cell responses correlate with disease outcomes, is incompletely addressed. Here, the authors state several positive correlations with effect ranges between 0.16 and 0.26. I wonder if the authors consider this a strong effect, given the high numbers of individuals included. As mentioned earlier, the reader is in need of an estimate what a mild or strong effect would be like. The topic of disease severity vs cellular immunity should also be discussed on findings in the above mentioned paper by Bacher et al. A novelty is the extensive correlation of CD8 responses with certain HLA alleles and the authors stress that this would provide useful information e.g. for vaccine design. Also on this background, it would also be interesting to correlate HLA haplotypes with disease severity to address if the identified alleles might contribute to protective responses as was e.g. already addressed for HLA-A2 (Habel, PNAS, 2020) that was associated with suboptimal CD8+ T cell responses.

As minor points,

- 1) FACS dot plots from donors more representative for the cohort should be chosen, more reflecting the differences between unexposed and infected individuals
- 2) it should be acknowledged and discussed that Miltenyi's S-Protein product does not encompass the complete S protein, but misses several parts in both S1 and S2 domain

Reviewer #2 (Remarks to the Author):

Olafsdottir et al present an analysis of T cell responses to SARS-CoV-2 antigens among a large population of confirmed SARS-CoV-2 cases or controls from Iceland. They find evidence for some cross-reactive responses to M and S proteins in the control samples, and robust T cell responses to all antigens in the case cohort. T cell responses increased with disease severity, age and antibody titres.

Overall, the authors should be commended for the amount of work performed - this study provides a rare opportunity to assess CD4 and CD8 T cell responses in a particularly large and well characterised cohort. Many of the results reported are consistent with the existing literature from other cohort studies. My main comment would be that the manuscript is quite dense, and due to the number of antigens studied, presents a large number of correlation plots that do not immediately present a clear message. Figure 2 could perhaps be moved to supplementary materials, and Figure 4 would benefit from either more labels/titles above the plots, or distinct figures for CD4 and CD8 T cells.

Main comments:

1. I don't understand the premise of the scatter plots in Fig 2 - are both cases and controls shown on the same plot? It seems unnecessary to include the controls on these plots in the responses were minimal relative to the cases. Also, it should be clarified for each plot whether the spearman correlation is significant (so that one does not rely on the 95% CI to interpret).
2. Why would there be a significant positive correlation between CMV responses and SARS-CoV-2 responses (Fig 2)? Does that imply that there are differences in general responsiveness to stimulation among participants?

3. The role for T cells in disease severity has been an ongoing topic of discussion - particularly as the authors assert that polyfunctional T cells are "higher quality", there should be further discussion of whether polyfunctionality was associated with severity in this cohort. I.e., did individuals with a more polyfunctional T cell response tend to have lower disease severity?

4. The conclusion that T cell responses are stable over time is in direct contrast to a number of published reports that have longitudinally tracked T cell frequencies. It is possible that the samples included in this study (particularly around 5-6 months post-infection) were sufficiently far into convalescence that responses had already declined to stable memory frequencies. In this case, the authors should re-word/modify their conclusions. Additionally, I notice the data in Fig S7, particularly the N-specific responses. Here it appears that there is a prominent decline in T cell responses from the ~100 day to ~200 day sample timepoints, especially compared to the CMV data. Can the authors comment on this? Is there a significant drop?

5. Discussion of the association between B*07:02 and N epitopes should include acknowledgement of multiple studies that have characterised these responses, including Peng et al Nat Immunol and Nguyen et al Immunity.

Minor comments:

1. Line 91 - incomplete sentence
2. Please include some measure of range/IQR/etc in Table 1 for the time from diagnosis row for the cases
3. Figure 4 title mentions sex, but no comparison by sex is included in that figure

Reviewer #3 (Remarks to the Author):

Brief summary of the manuscript

The paper provides new insights on the impact of HLA class I / Ib and class II genotype on age-related T-cell responses following COVID-19. The authors showed that T-cell responses were stable up to eight months after SARS-CoV-2 infection, irrespective of disease severity. CD4+ T-cell responses correlated with disease severity, humoral responses and age, while CD8+ T-cell responses correlated with age and functional antibodies. CD8+ T-cell responses associated with several class I HLA alleles whereas CD4+ T-cell responses were less associated with specific class II HLA alleles.

Overall impression of the work

From my point of view, the main limitation of the study is the narrow age distribution in SARS-CoV-2 group. All of them are below 60 years old. Another concern is units at the figures with flow cytometry data. In Dan et al. [10.1126/science.abf4063] SARS-CoV-2-specific CD8+ T cells level rangers between 0.1 and 1 %. On the Fig. S7 in supplementary file of present work SARS-CoV-2-specific CD8+ T cells level rangers between 0.001 and 0.0001%. What explains such a great disparity?

Also, the paper needs thorough editing and proof-reading, preferably by a native speaker.

Reviewers' comments:

Reviewer #1 (Remarks to the Author):

In their manuscript, Olafsdottir et al. analyze CD4 and CD8 T cell responses in healthy controls and SARS-CoV-2 infected individuals. Frequencies of antigen-specific T cells are determined based on cytokine production after stimulation with 4 CoV-2 proteins (M, N, S, S1 domain of S) and CMV pp65 as control. Results are correlated to age, sex and COVID-19 disease severity. Furthermore, based on HLA haplotype analysis, associations between CD8 responses and MHC-I gene expression are studied. The authors end with longitudinal analyses, addressing how T cell immunity is maintained or altered months after infection. Clearly, the main selling point of the manuscript is the number of individuals included, exceeding by far most, but not all studies focusing on FACS based T cell analysis. Also the correlation of HLA haplotypes and CD8 responses has not been conducted in such depth. SARS-CoV2 related studies have fueled interest in large cohorts that are studied in great detail; in that respect, Bacher et al. extensively analyzed 120 individuals (Bacher, P, Immunity, 2020), while Loyal et al. included more than 500 individuals for CoV2-specific T cell analysis, including MHC-II haplotype analysis (Loyal, L, Science, 2021).

We thank the reviewer for the constructive comments. Please see our specific answers below each comment. Text directly from the manuscript is written in *italics* and we have underlined additions made to the manuscript

There are several main points of concern regarding the quality of the manuscript:

1) The main focus of the study is the analysis of multi- or polyfunctional T cells. The authors state that on average, 68k CD4+ T cells and 32k CD8+ T cells were acquired. Looking e.g. at Fig. 1C, mean frequencies of polyfunctional CD4+ T cells are usually below 1 in 10.000 – meaning that on average, less than 6.8 cells fall in this category. Based on this, it is doubtful if the assay design regarding starting cell numbers will adequately allow to quantify these very rare cells and ensuing conclusions and correlation analyses in Fig. 2. This problem becomes even more obvious for CD8+ T cells, where many responses are between 1 in 10.000 and 1 in 100.000. Since on average, only 32k cells were analyzed, this results in 3.2 to 0.32 specific cells.

The reviewer is correct in that the polyfunctional cells (secreting all three cytokines) are few. However, we want to point out that the main conclusions of this paper are not only based on the polyfunctional cells, for both CD4+ and CD8+ T-cells we also analyze the correlation of single and double cytokine secreting cells that all support the same pattern as observed for the polyfunctional cells (Figure 1C-D, Supplementary figures 1-4 and supplementary tables 2-3). This is clearly stated throughout the results chapters.

We note in the manuscript that CD8+ T cell responses are lower than CD4+ T cell responses, lines 325-327: *“The CD8+ T-cell responses to all four SARS-CoV-2 proteins were also*

significantly higher in the cases than controls, although more modest than CD4⁺ T-cell responses, consistent with other studies". Therefore, we chose to base our main analysis of CD8⁺ T cells on all IFN γ secreting cells, although we analyze also other, double and triple cytokine producing cells.

We further want to draw the attention of the reviewer to Figure 2 (now Figure S2 as it was moved to supplements). The presence of a correlation among cases of triple cytokine producing CD4⁺ T-cells convinced us that, although we are measuring few polyfunctional CD4⁺ T-cells, they reflect a true reaction to the SARS-CoV-2 antigens. This was unlike the low correlation between SARS-CoV-2 antigens and the CMV peptide pool. Further, these correlations are not present among uninfected individuals, except for the correlation between M and S. Based on these results, polyfunctional SARS-CoV-2 CD4⁺ T-cell responses should not be deemed as inadequately quantified and dismissed.

The methods part does not state how many cells were used per stimulation; if the mentioned 1x10⁷ were used per stimulation condition, recovery after thawing would be extremely poor.

We have clarified that for each stimulation condition we used 200,000 cells which has been added to the method section:

Line 585 *"Cells were split into 2x10⁵ cells per well and stimulated with 0.5 μ g/mL SARS-CoV-2 Prot_M, _N, _S, _S1 or CMV peptides pools along with 0.3 μ g/mL CD28 and CD49d antibodies co-stimulation for 8h total, 5 μ g/mL Brefeldin A and Monensin were added at 5 μ g/mL after 30 min."*

2) What is still incompletely understood is the relation of immune parameters and disease severity. This aspect, together with the influence of age, is addressed in Fig. 4. Since leuko- or lymphopenia has been identified as a hallmark of severe COVID, it would be helpful not only to address frequencies of antigen-specific T cells, but also absolute counts, providing information on whether higher disease score patients harbor also higher absolute antigen specific T cell numbers.

We appreciate the point raised by the reviewer that increased frequency of antigen specific T-cells observed in higher disease score patients, might be affected by a potential difference in the total number of T-cells in these patients. Following the reviewer's suggestion, we have performed additional analysis on the association with absolute cell counts of SARS-CoV-2 reactive T-cell responses using a quasi-Poisson regression model. These analysis have been added into Table S2 for CD4⁺ T-cells and Table S3 for CD8⁺ T-cell responses. Overall, the association observed when testing the frequency of SARS-CoV-2 was reflected in the association with the absolute cell count and the effect was always in the same direction using the two parameters, meaning that not only do those that got more severe disease have higher frequency of SARS-CoV-2 reactive T-cells but also higher absolute count. We interpret this as a result of a more abundant and prolonged antigen stimulation during the acute phase.

We have added a paragraph on this, lines 365-371: “Further, the increase in frequency of antigen-specific T-cells in higher disease score patients was not explained by prolonged lymphopenia as was described in the acute phase of COVID⁴² as we observed similar results when testing for association with absolute counts of antigen-reactive T-cells (Table S2). Therefore, the association between increased CD4⁺ T cell responses in convalescent individuals and disease severity likely reflects more abundant and prolonged antigen stimulation and consequent expansion of the adaptive immune response during the acute phase of the infection.”

Also data is missing on how a positive correlation between CD4+ T cell numbers and severity is addressed; the text p12, l219 states a positive correlation e.g. in Fig 4C, but no test is mentioned or illustrated here.

We performed two types of statistical analysis as explained in the methods chapter lines 688-692 “We estimated correlation between T-cell responses and other variables using Spearman’s correlation coefficient and used a Jackknife method to calculate confidence interval⁵⁰. When testing for association between T-cell responses we performed linear regression, using the logarithm of T-cell responses as a response and using age and sex as covariates – except when testing for association with these variables.”

We apologize that it was not always clear in the results chapter which analysis was performed. We have added the following sentences for clarification

Lines 111-118 “We performed linear regression to test for association between T-cell responses in different groups and report effects that represent differences in the logarithm of the T-cell responses, either as frequency of cytokine secreting cells out of all CD4⁺ T-cells or absolute count of CD4⁺ cytokine secreting T-cells (Table S2 and corresponding information for CD8⁺ T-cell responses in Table S3). For simplification, we only refer to the effects on frequency of T-cell responses in the text and give effect range (ranging from the lowest to highest effect), when referring to multiple cell sub-populations and/or different stimulations.”

Line 134 “We estimated correlation between T-cell responses with Spearman’s correlation coefficient (ρ).”

Line 150 we added what is underlined: “Cases had higher frequencies than controls of IFN- γ CD8⁺ T-cells responding to all four proteins as estimated by linear regression (effect range 0.20 to 1.12, **Figure 3** and **Table S3**).”

Line 208-210: “We analyzed the association between age and T-cell responses with linear regression by splitting cases and controls into three age bins (18-40, 41-60 and 61-91 years of age) and report differences in the logarithm of the T-cell responses, per age bin”.

Line 226-228 “Association between T-cell responses and disease severity was tested by linear regression with effects representing differences in the logarithm of the T-cell responses in cases, per unit of the severity scale.”

Line 231 Instead of referring to positive correlation we have changed the sentence to “CD4⁺ T-cell responses against all four proteins increased with each unit of disease severity”

The following sentence was added to the figure legends of Figure 1 (line 444-445), 2 (line 467) and 3 (line 493-494) “Association between T-cell responses in different groups was tested by linear regression”

The following sentence was added to the figure legends of Figure 4 (line 503-505) “Correlation between T-cell responses was estimated using Spearman’s correlation coefficient and a Jackknife method was used to calculate confidence intervals.”

The following sentences was added to figure legend 5 (line 586-589) “ The correlation between T-cell responses and functional antibody responses shown in A and B was estimated using Spearman’s correlation coefficient and a Jackknife method was used to calculate confidence intervals.”

3) Overall, it is not easy to follow statistical analyses as presented in the paper and judge if the reported results are meaningful. For instance, effect ranges or, alternatively, effects are calculated without extensively naming and explaining the methods for an unexperienced reader. Supporting information is missing allowing basic estimates for weak vs strong effects or ranges. Inter alia, the reported effects (for example starting from line164) and effect ranges (for example from line147) need to be explained more clearly and names of the test system included. It is also not clear how the adjusted analysis (line687) is performed - a linear model is described but 'sex' as a non-parametric variable listened - was a glm used? There are no R²/adjR² and DFs mentioned. In addition, it is not reported that the requirements of the used methods are fulfilled (such as homoscedasticity for the multivariate model). Overall, the statistical methods part needs to be improved and elaborated.

To simplify the presentation of the results, we have converted the effect estimates from the genetics association analysis to the logarithm of T-cell response scale. This is the scale that all other association results in the paper are presented on and hopefully this change will improve the readability of the paper.

We have added details to the „Statistical analysis“ section that explain more clearly the statistical analysis that we performed. We have added adjusted R² estimates to the Supplementary Tables with linear regression results. To assess the robustness of our linear regression results we have added analysis using general least squares (gls) models where we allowed the variance to depend linearly on each of the covariates. The gls results are shown in Supplementary Tables 2 and 4. Effect estimates and significance levels we obtained for linear regression and gls were comparable and did not alter our inference.

We have added a description of how the association between T-cell responses and HLA were calculated both in the Results chapter and with further details in the methods chapter.

Line 167-168: “We used linear regression to test for association of T-cell responses with HLA genotype and reported the effects as changes in the logarithm of T-cell responses”

Line 692-710 “ Sex was included as an indicator variable (0 for males and 1 for females). Significance was assessed with the standard linear regression t-test with $n-p$ degrees of freedom, where n is the number of available observations and p is the number of parameters (2 or 3). The adjusted R2 for each linear model is shown in the supplementary tables.

Since T-cell responses are highly skewed and their variance depends on the T-cell response itself, we performed additional analysis using general least squares models (as implemented in the `gls` function in R) where we allowed the variance to depend linearly on each of the covariates included in the model using the `varFixed()` option. Overall, the results did not differ substantially between the simple linear regression and general least squares analysis.

We also test for association with the absolute number of responding T-cells, without dividing by the number of CD4+ or CD8+ cells, using the quasi-Poisson model implemented in the `glm` R function. As for the linear regression test, we used age and sex as covariates.

In the genetic association analysis, where we associated T-cell responses with HLA genotypes, we used inverse normal standardization to convert the T-cell responses to a standard normal distribution and then performed standard genetic association testing as previously described⁵³. To convert effect estimates from standard deviations to the logarithm of T-cell response, we multiplied the effect estimates in standard deviations with the standard deviations of logarithm of T-cell responses.”

Regarding the figures, they should be reduced in number and be summarized. For an example, the boxplots on P. 48 - 49 could be merged together. In addition, the presentation in age ribbons appears not meaningful since and a metric scale would allow a better illustration of data.

We have made the following changes to the figures in the manuscript:

1. Moved figure 2 to supplements
2. Moved Figure 4A - 4C to supplements
3. Removed Figure 5A and 5B and only refer to figure S8 for the correlation between T-cell responses and pan Ig responses.
4. Added CD4 and CD8 labels to figures 3,4 and 5

4) Given that T cell responses have been shown to depend on patient predisposition, it would be helpful to correlate responses with comorbidities and not only with age; this information will likely be available for the cohort and greatly enrich pioneering studies as the one from Sattler et al. (Sattler, JCI, 2020) that might also serve as a basis for discussion of this aspect.

Upon the request from the reviewer, we tested association between T-cell responses and history of having been diagnosed with any of the following comorbidities: asthma, cancer, type 2 diabetes, coronary artery disease, hypertension, and severe obesity (BMI > 35). No comorbidity associated with a T-cell response after accounting for multiple testing. We have added a supplementary table S8 showing the tested associations and added the following sentence in the results chapter line 244-246 *“No significant association was observed between T-cell responses and history of having been diagnosed with any of the major COVID19 comorbidities following adjustment for multiple testing (Table S8)”*

5) In summary, it appears that the authors also see their study as a reproduction of existing studies with a larger cohort. This is reflected by that fact that each of their summary sentences in the 1st paragraph of the discussion (p16) ends with references stating similar findings. This applies to T cell targets, CD4/CD8 response characteristics, longitudinal follow up and correlations of T cell with antibody responses. One of the most important questions, namely, how T cell responses correlate with disease outcomes, is incompletely addressed. Here, the authors state several positive correlations with effect ranges between 0.16 and 0.26. I wonder if the authors consider this a strong effect, given the high numbers of individuals included. As mentioned earlier, the reader is in need of an estimate what a mild or strong effect would be like. The topic of disease severity vs cellular immunity should also be discussed on findings in the above mentioned paper by Bacher et al.

The comparison part to other studies was certainly not the main aim of our study but while analyzing and writing up our data, several papers were published on the subject and we have acknowledged and tried to compare what has already been reported in smaller studies with our large cohort. It is true that the extensive monitoring service of COVID-19 patients in Iceland allowed us to classify a large fraction (71%) of our cases into four different severity categories and correlate with T-cell responses.

We admit that it can be difficult to interpret the effect sizes as strong or weak and we thank the reviewer for giving us the chance to clarify this. The effect sizes always represent differences in the logarithm of the T-cell responses observed between any given groups that are being compared. For the disease severity the effect sizes therefore, represent differences in the logarithm of the frequency of stimulated cells from cases, per unit of the severity scale. In order to put this into context, we have added a comparison of the effect sizes observed for the severity with the effect sizes observed when comparing T-cell responses between cases and controls. For example, we observed an effect range of 0.66 to 1.09 for the different SARS-CoV-2 peptides, when comparing polyfunctional CD4⁺ T-cell responses between cases and controls, i.e. cases have on average 4.6 to 10-fold higher SARS-CoV-2 reactive CD4⁺ T-cell responses than controls. When we then look at the effect sizes observed for the disease severity, we observed effect sizes ranging from 0.12 to 0.26, meaning that on average, SARS-CoV-2 reactive CD4⁺ T-cell responses increase 1.3 to 1.8 fold for each increase in severity category. When we then compare those with the mildest infection (severity category 1) with controls we observe effect sizes ranging from 0.48 to

0.80 (i.e. mildest cases have on average 3 to 6.3 fold higher polyfunctional CD4⁺ T-cells compared with controls). Therefore, it is clear that although CD4⁺ T-cell responses significantly increase with disease severity, the increase is considerably smaller than when comparing cases to controls or even when comparing those that got the mildest form of the disease with controls.

The following text has been added to the result chapter for clarification (in addition of what was already mentioned above):

1. Line 226-228 "Association between T-cell responses and disease severity was tested by linear regression with effects representing differences in the logarithm of the fraction of stimulated cells from cases, per unit of the severity scale."
2. Line 234-240 "Putting this into context with increased T-cell responses observed in cases versus controls, we observe that although CD4⁺ T-cell responses significantly increase with disease severity of the four groups of cytokine producing cells mentioned above (effect range 0.12 to 0.27), the increase is considerably smaller than when comparing cases to controls (effect range 0.66 to 1.56) or even when comparing those that got the mildest form of the disease with controls (effect range 0.48 to 1.31) (Table S2)."
3. We have added the following paragraph on the mentioned paper by Bacher *et al.* to our discussion part line 361-371 "We cannot exclude that the association of increased CD4⁺ T-cell responses with increased severity and age is due to polyclonal and low avidity T-cells as reported by Bacher *et al.*⁴¹ although we observed significant increase in double and triple cytokine producing cells, indicating high functionality of those cells. Further, the increase in frequency of antigen-specific T-cells in higher disease score patients was not explained by prolonged lymphopenia as was described in the acute phase of COVID⁴² as we observed similar results when testing for association with absolute counts of antigen-reactive T-cells (Table S2). Therefore, the association between increased CD4⁺ T cell responses in convalescent individuals and disease severity likely reflects more abundant and prolonged antigen stimulation and consequent expansion of the adaptive immune response during the acute phase of the infection."

6) A novelty is the extensive correlation of CD8 responses with certain HLA alleles and the authors stress that this would provide useful information e.g. for vaccine design. Also on this background, it would also be interesting to correlate HLA haplotypes with disease severity to address if the identified alleles might contribute to protective responses as was e.g. already addressed for HLA-A2 (Habel, PNAS, 2020) that was associated with suboptimal CD8⁺ T cell responses.

This is a very good point and was performed upon request of the reviewer. None of the HLA alleles that associated with T-cell responses were found to associate with disease severity following correction for multiple testing ($P=0.05/5=0.01$). The only HLA allele that associated nominally with increased disease severity was HLA-A*01:01 ($P=0.044$) Effect=0.118, that we also found to be associated with decreased S1 reactive CD8⁺ T-cell responses.

We have added this information as supplementary Table S5 and added the following text to the manuscript line 188-189 “ *No significant association was observed between T-cell associating HLA alleles and disease severity when correcting for multiple testing (Table S5).*”

As minor points,

1) FACS dot plots from donors more representative for the cohort should be chosen, more reflecting the differences between unexposed and infected individuals

We spent quite some time on this and we choose to keep the FACS plots as they are since they are representative of our cohort, rather than exemplifying where we have the greatest difference between unexposed and infected individual.

2) it should be acknowledged and discussed that Miltenyi’s S-Protein product does not encompass the complete S protein, but misses several parts in both S1 and S2 domain

We have clarified this in the discussion line 322-324 “*We observed robust CD4+ T-cell responses following SARS-CoV-2 infection upon stimulation with selected domains of the S protein (see methods for details),*”

And in the methods chapter line 587-589: “*We note that only selected sequence domains (aa 304-338, 421-475, 492-519, 683-707, 741-770, 785-802, and 885 – 1273) are included in the S peptide stimulation pool.*”

Reviewer #2 (Remarks to the Author):

Olafsdottir et al present an analysis of T cell responses to SARS-CoV-2 antigens among a large population of confirmed SARS-CoV-2 cases or controls from Iceland. They find evidence for some cross-reactive responses to M and S proteins in the control samples, and robust T cell responses to all antigens in the case cohort. T cell responses increased with disease severity, age and antibody titres.

Overall, the authors should be commended for the amount of work performed - this study provides a rare opportunity to assess CD4 and CD8 T cell responses in a particularly large and well characterised cohort. Many of the results reported are consistent with the existing literature from other cohort studies. My main comment would be that the manuscript is quite dense, and due to the number of antigens studied, presents a large number of correlation plots that do not immediately present a clear message. Figure 2 could perhaps be moved to supplementary materials, and Figure 4 would benefit from either more labels/titles above the plots, or distinct figures for CD4 and CD8 T cells.

We thank the reviewer for the constructive comments. Please see our specific answers below each comment. Text directly from the manuscript is written in *italics* and we have added additions made to the manuscript.

We agree that the manuscript is dense but we find it important to make full use of our large cohort.

However, we have made some changes in order to simplify the manuscript for the reader including:

- 1 Moved figure 2 to supplements as suggested by the reviewer
- 2 Shortened the chapter on correlation between CD4⁺ and CD8⁺ T-cell responses and moved it to line 258-264 in the chapter on *“SARS-CoV-2 reactive CD4+ T-cell responses correlated with age, sex and disease severity while CD8+ T-cell responses only correlated with age”*
- 3 Added CD4 and CD8 labels to figures 3, 5 and 5 as suggested by the reviewer
- 4 Moved Figure 4A - 4C to supplements
- 5 Removed Figure 5A and 5B and only refer to figure S8 for the correlation between T-cell responses and pan Ig responses.

Main comments:

1. I don't understand the premise of the scatter plots in Fig 2 - are both cases and controls shown on the same plot? It seems unnecessary to include the controls on these plots in the responses were minimal relative to the cases. Also, it should be clarified for each plot whether the spearman correlation is significant (so that one does not rely on the 95% CI to interpret).

The point of including the controls in the figure is to demonstrate the difference between the cases and the controls. Indeed, there is very little response in the controls. In addition to the confidence intervals of the Spearman correlations we also plot confidence intervals for the regression lines which provide a clear visual indication of when the correlation are significantly different from 0.

2. Why would there be a significant positive correlation between CMV responses and SARS-CoV-2 responses (Fig 2)? Does that imply that there are differences in general responsiveness to stimulation among participants?

This is an excellent point. It does certainly indicate that, although we also want to point out that the correlation between CMV and each of the SARS-CoV2 antigen is lower than what is observed between paired SARS-CoV-2 antigen in those that got the infection. We have added a sentence on this in line 144-147 *“Further, we note that there is low but significant correlation between T-cell responses against each of the SARS-CoV-2 and the CMV peptides indicating that there is individual difference in memory T-cell responses among the participants in the study (Figure 2 and Figure S2-S3).”*

3. The role for T cells in disease severity has been an ongoing topic of discussion - particularly as the authors assert that polyfunctional T cells are "higher quality", there should be further discussion of whether polyfunctionality was associated with severity in

this cohort. I.e., did individuals with a more polyfunctional T cell response tend to have lower disease severity?

Actually quite the opposite, that is those with most severe disease also had the highest CD4⁺ T-cell responses (including polyfunctional T-cells) which is also observed for the antibody response. Here we need to keep in mind that the samples are taken 3 to 8 months following infection and therefore the high memory T-cell responses likely reflect a stronger antigen stimulation and expansion during the acute phase of severe infection rather than a direct relationship between polyfunctional T-cell responses and worse outcome. Following a request from reviewer #1 we also performed the association analysis between severity and absolute SARS-CoV-2 reactive T-cell count that revealed similar results. We added the following text to the discussion line 361-371

“ We cannot exclude that the association of increased CD4⁺ T-cell responses with increased severity and age is due to polyclonal and low avidity T-cells as reported by Bacher et al ⁴¹ although we observed significant increase in double and triple cytokine producing cells, indicating high functionality of those cells. Further, the increase in frequency of antigen-specific T-cells in higher disease score patients was not explained by prolonged lymphopenia as was described in the acute phase of COVID⁴² as we observed similar results when testing for association with absolute counts of antigen-reactive T-cells (Table S2). Therefore, the association between increased CD4⁺ T cell responses in convalescent individuals and disease severity likely reflects more abundant and prolonged antigen stimulation and consequent expansion of the adaptive immune response during the acute phase of the infection.”

4. The conclusion that T cell responses are stable over time is in direct contrast to a number of published reports that have longitudinally tracked T cell frequencies. It is possible that the samples included in this study (particularly around 5-6 months post-infection) were sufficiently far into convalescence that responses had already declined to stable memory frequencies. In this case, the authors should re-word/modify their conclusions. Additionally, I notice the data in Fig S7, particularly the N-specific responses. Here it appears that there is a prominent decline in T cell responses from the ~100 day to ~200 day sample timepoints, especially compared to the CMV data. Can the authors comment on this? Is there a significant drop?

The reviewer is correct, that it is possible that a significant drop in T-cell frequency has happened in the first three months (not included in our analysis). We have re-phrased our conclusion accordingly on line 330 *“Our analysis revealed relatively stable SARS-CoV-2 reactive CD4⁺ and CD8⁺ T-cell responses during the time-frame of our study (i.e. three to eight months from SARS-CoV-2 infection), with no indication of decline during this period, in agreement with previous reports ^{5,26}.”*

The drops in figure S7 are not statistically significant. The number of individuals that we have serial samples from is small (only 30). So not much statistical inference can be based on this dataset.

5. Discussion of the association between B*07:02 and N epitopes should include acknowledgement of multiple studies that have characterised these responses, including Peng et al Nat Immunol and Nguyen et al Immunity.

We thank the reviewer for the opportunity to add this to our discussion. We have changed our discussion on this based on the recommendations from the reviewer line 337-345 “ ~~In line with our results peptides from the N protein presented in HLA-B*07:02 have been shown to be recognized by a large fraction of CD8+ T cells in convalescent individuals when stained with tetramers directly ex vivo~~²⁷. We observed that many of the N-derived peptides predicted to bind to HLA-B*07:02 are well conserved between SARS-CoV-2 and other members of the Coronaviridae. *In line with our results, several recent papers have reported immunodominant N derived peptides when presented by the HLA-B*07:02 allele*²⁷⁻³⁰*and cytotoxic T-cell response directed against N*₁₀₅₋₁₁₃*-B*07:02 was associated with less severe COVID-19 disease. Interestingly, high frequency of naïve CD8+ T-cell precursors recognizing N*₁₀₅₋₁₁₃*-HLA-B*07:02 was also found in pre-pandemic samples*^{30,31}. It is therefore tempting to speculate that the strong N reactive CD8+ T-cell responses observed in carriers of HLA-B*07:02 in our nationwide study population reflects an expansion ~~booster response~~ of pre-existing CD8+ T-cell immunity ~~against HCoVs cross-reactive CD8+ T-cells.~~”

Minor comments:

- 1. Line 91 - incomplete sentence** We have fixed this by removing the word „as“
- 2. Please include some measure of range/IQR/etc in Table 1 for the time from diagnosis row for the cases**

As suggested we have added the first and third quartiles to the table. Unfortunately, we made the mistake of displaying the mean rather than the median in the table. This error has been corrected.

- 3. Figure 4 title mentions sex, but no comparison by sex is included in that figure.** We have corrected this, comparison by sex is included in Table S6.

Reviewer #3 (Remarks to the Author):

Brief summary of the manuscript

The paper provides new insights on the impact of HLA class I / Ib and class II genotype on age-related T-cell responses following COVID-19. The authors showed that T-cell responses were stable up to eight months after SARS-CoV-2 infection, irrespective of disease severity. CD4+ T-cell responses correlated with disease severity, humoral responses and age, while CD8+ T-cell responses correlated with age and functional antibodies. CD8+ T-cell responses associated with several class I HLA alleles whereas CD4+ T-cell responses were less associated with specific class II HLA alleles.

We thank the reviewer for the constructive comments. Please see our specific answers below each comment. Text directly from the manuscript is written in *italics* and we have

underlined additions made to the manuscript.

Overall impression of the work

From my point of view, the main limitation of the study is the narrow age distribution in SARS-CoV-2 group. All of them are below 60 years old.

There is some misunderstanding here as we also have individuals between 60-90 years of age as can be seen in figure 4.

Another concern is units at the figures with flow cytometry data. In Dan et al.

[10.1126/science.abf4063] SARS-CoV-2-specific CD8+ T cells level rangers between 0.1 and 1 %. On the Fig. S7 in supplementary file of present work SARS-CoV-2-specific CD8+ T cells level rangers between 0.001 and 0.0001%. What explains such a great disparity?

The disparity is likely explained by the different experimental setup as well as readouts used in our study compared with Dan et al. We chose to use a short *in vitro* stimulation of 8h and stain for cytokine producing cells with Intracellular-cytokine staining (ICS). In contrast, Dan et al estimated the SARS-CoV-2 reactive T-cell responses by staining for surface markers of activated T-cells, Activation-induced markers (AIM), following 24h stimulation. It is well documented that AIM detects broader T-cell response than ICS as it detects the total pool of antigen-reactive T-cell responses PMID: [30065162](https://pubmed.ncbi.nlm.nih.gov/30065162/). However, ICS better reflects the functionality of the T-cell response and here we chose to focus on the Th1 response that is well established to provide protection against viral infections as well as we opted for a short *in vitro* stimulation protocol to minimize artifacts that may arise in a prolonged *in vitro* cultivation of the primary blood cells. We are therefore aware that we are estimating the highly functional anti-viral response at the cost of underestimating the total SARS-CoV-2 reactive T-cell response. We have emphasized our reasons for choosing the ICS in the introduction line 85-86 „ *We studied functional anti-viral immunity by assessing CD4⁺ and CD8⁺ T-cells secreting the canonical type 1 cytokines: IFN- γ , TNF- α and IL-2 upon stimulation with SARS-CoV-2 proteins¹⁸*”

Also, the paper needs thorough editing and proof-reading, preferably by a native speaker.

We have made extensive editing of the text that we hope is acceptable to the reviewer. All of the changes made can be observed in track changes version of the manuscript.

Changes in figures:

Figure 2 was moved to supplements and is now found as figure S2.

Moved Figure 4A - 4C to supplements that are now found in figure S7.

Removed Figure 5A and 5B and refer to figure S8 for the correlation between T-cell responses and pan Ig responses.

Subsequently all main and supplementary figures were renumbered.

Added CD4 and CD8 labels to figures 3, 5 and 5 as suggested by the reviewer

The changed figures are shown in the following pages

Figure 3. SARS-CoV-2 reactive CD4⁺ T-cell responses correlate with age and disease severity while CD8⁺ T-cell responses only correlate with age. Box plots showing correlation between SARS-CoV-2 reactive (A) polyfunctional CD4⁺ as well as (B) all IFN- γ ⁺ CD8⁺ T-cells and age. Frequency of cytokine secreting T-cells (as indicated on the Y-axis) and age (that was split into age bins as indicated on the X-axis) following different stimulations (as indicated above each plot). Uninfected controls are displayed in red and SARS-CoV-2 cases in blue. (C) Box plots show correlation of polyfunctional CD4⁺ T-cell responses (Y axis) with disease severity (indicated by the different colors) upon different stimulation conditions (X axes). For comparison, we show the response induced by each stimulation condition in the uninfected controls (ctrl). Frequency of polyfunctional CD4⁺ T-cell responses in SARS-CoV-2 infected individuals were grouped into four severity groups based on data from questionnaire and/or medical records. 1: Asymptomatic/mild (N=207); 2: moderate (N=139); 3: severe (N=163); 4: hospitalized (N=30) (B). The bottom and top of the boxes correspond to the 25th (Q1) and 75th (Q3) percentiles, the line inside the box corresponds to the median, and the whiskers are located at $\max(\min(\text{Expression}), Q1 - 1.5 \text{ IQR})$ and $\min(\max(\text{Expression}), Q3 + 1.5 \text{ IQR})$, respectively (where IQR is the interquartile range = $Q3 - Q1$). (D) Estimates for differences in the logarithm of the percentage of (D) polyfunctional CD4⁺ T-cells out of all CD4⁺ T-cell responding to the different stimulation conditions between unit of the severity scale of the cases and same is shown for (E) all IFN- γ ⁺ CD8⁺ T-cells. (F) Difference in the logarithm of the percentage of polyfunctional CD4⁺ T-cells out of all CD4⁺ T-cell responding to the different

stimulation conditions between cases with the mildest disease severity and uninfected controls and the same is shown for (G) CD8⁺ T-cells. 95% confidence intervals (CI) are shown. See also Tables S2 and S3. Association between T-cell responses in different groups was tested by linear regression

Figure 4. SARS-CoV-2 induced T-cell responses do not decline in the first eight months after diagnosis. Cross-sectional analysis of SARS-CoV-2 reactive (A) polyfunctional CD4⁺ and (B) all IFN- γ ⁺ CD8⁺ T-cells from all subjects in the study where samples (N=759) were collected up to 259 days from diagnosis. Y-axes shows percentage of polyfunctional (A) CD4⁺ and (B) CD8⁺ T-cells responding to the stimulation indicated on the Y axes (U, CMV, M, N, S or S1). Y-axis indicates time from diagnosis in days. Correlation between T-cell responses was estimated using Spearman's correlation coefficient and a Jackknife method was used to calculate confidence intervals.

Figure S7. Kinetics of the SARS-CoV-2 induced T-cell responses in the first eight months after diagnosis. (A) Overview of 90 paired samples from the same individuals collected 3-6 months from SARS-CoV-2 diagnosis (sample 1) and again 3-5 months later (sample 2). Correlation between sample 1 and sample 2 for frequency of (B) polyfunctional (IFN- γ ⁺TNF- α ⁺IL-2⁺) SARS-CoV-2 reactive CD4⁺ T-cells and (C) all IFN- γ ⁺ CD8⁺ T-cells. Cross-sectional analysis of SARS-CoV-2 reactive (A) polyfunctional CD4⁺ and (B) all IFN- γ ⁺ CD8⁺ T-cells from all subjects in the study where samples (N=759) were collected up to 259 days from diagnosis.

Figure 5. SARS-CoV-2 reactive CD4⁺ and to lesser extent CD8⁺ T-cell responses correlate with antibody levels. Scatter plot show correlation between (A) polyfunctional CD4⁺ as well as (B) all IFN- γ ⁺ CD8⁺ T-cells and functional antibody responses. Inhibition of the antibodies was measured using an MSD-ECLIA based ACE2 blocking assay and reflects how well the plasma antibodies inhibit binding between soluble ACE2 and plate bound S1-RBD. Inhibition efficacy of the antibodies is given on log scale on the X-axes and percentage (A) polyfunctional CD4⁺ T-cells out of all CD4⁺ T-cells and (B) all IFN- γ ⁺ CD8⁺ T-cells out of all CD8⁺ T-cells are given on the Y-axes. The correlation between T-cell responses and functional antibody responses shown in A and B was estimated using Spearman's correlation coefficient and a Jackknife method was used to calculate confidence intervals.

Reviewers' comments:

Reviewer #1 (Remarks to the Author):

The authors provide an extensive piece of work based on the reviewers comments and critiques which has to be highly acknowledged. The paper nevertheless lacks in my opinion, apart from the tremendous number of individuals included, a substantial step towards a better understanding of the relation of T cell immunity and SARS-CoV2 infection outcome that separates it from the plethora of existing studies. In addition, despite the re-organization of the manuscript, a main focus is still on polyfunctionality, especially in Fig.3 where the important topic of associations with disease severity is addressed; as mentioned in the first review, this has to be seen critical due to low cell numbers. In addition, after carefully looking at the automatic gating strategy exemplarily shown in Suppl. S13, gates cut through the negative population, leading to high frequencies of false positive cells (e.g. 1.09% of IFN γ single positive CD4+ cells from which the majority is false positive). Gates look way different in Fig. 1 and 2, leaving the reviewer insecure how specific T cells were quantified overall.

Reviewer #2 (Remarks to the Author):

The authors have responded to the reviewers' concerns, and revised the manuscript appropriately to improve its clarity. I have no further comments

Reviewer #3 (Remarks to the Author):

The authors have improved the overall presentation of the paper, as suggested by the Reviewers.

The Material and Method section has been improved.

The results obtained from this work are sound and important particularly in the context of Covid-19 pandemic that we all have faced.

Reviewer #1 (Remarks to the Author):

The authors provide an extensive piece of work based on the reviewers comments and critiques which has to be highly acknowledged. The paper nevertheless lacks in my opinion, apart from the tremendous number of individuals included, a substantial step towards a better understanding of the relation of T cell immunity and SARS-CoV2 infection outcome that separates it from the plethora of existing studies.

We are very sorry to hear that the reviewer does not feel that our work has further extended his/her knowledge on the heterogeneity of SARS-CoV-2 reactive T-cell responses. In our view the main strength of the study is the large number of samples run within a well-defined study population that makes us well powered to robustly study demographic and clinical correlates of T-cell responses e.g. age, sex and disease severity. The reviewer is correct that there is a plethora of existing studies, many of which are based on small numbers of individuals, with heterogenous experimental conditions and readouts. This has resulted in many contradictory findings that have been reported that are certainly worth testing in a well powered study such as ours. In this manuscript we provide functional readout of T-cells secreting the canonical anti-viral cytokines IFN γ , TNF α and IL2 in a large cohort making this one of very few, if not the only nation-wide study, on SARS-CoV-2 reactive T-cell responses. A couple of recently published reviews agree with the point that a well-powered study on SARS-CoV-2 reactive T-cell responses is lacking. (Vardhana et. al, Science Immunology, 2022 PMID: 35324269, Paul Moss, Nature Immunology, 2022, PMID:35105982). Further, we want to point out that although several papers have reported on HLA association with disease severity, many of those are based on very small number of cases, meaning that only the most common HLA could be tested. Here, we test directly the association between all HLA alleles found in Icelanders with frequency above 0.1% and T-cell responses (resulting in total of 269 four digit HLA alleles being tested). We therefore, believe that this is the most comprehensive study on the association between all different HLA alleles and SARS-CoV-2 reactive T-cell responses following infection. Taking together, we strongly feel that our results provide valuable insight into the T-cell mediated immunity following SARS-CoV-2 infection.

In addition, despite the re-organization of the manuscript, a main focus is still on polyfunctionality, especially in Fig.3 where the important topic of associations with disease severity is addressed; as mentioned in the first review, this has to be seen critical due to low cell numbers.

As explained in our response to the initial concerns of the reviewer, we do not agree with the reviewer that our analysis of the polyfunctional CD4⁺ T-cell response is inadequately quantified. From the original response *"We further want to draw the attention of the reviewer to Figure 2 (now Figure S2 as it was moved to supplements), since this correlation of triple cytokine producing CD4⁺ T-cells observed between all stimulations in cases convinced us that although we are measuring few polyfunctional CD4⁺ T-cells, they are reflecting a true reaction to the SARS-CoV-2 antigens as compared with the low correlation observed between any SARS-CoV-2 antigens and the CMV peptide pool. Further, these correlations are not observed in uninfected individuals except for correlation between M and S responding CD4⁺ T-cells. Based on these data combined, polyfunctional SARS-CoV-2 CD4⁺ T-cell responses should not be deemed as inadequately quantified and dismissed."*

It is clear from the text in the manuscript that although, we chose to show polyfunctional CD4⁺ T-cells as representatives for the severity correlation (due to their previously reported correlation with protection against intracellular pathogens), we do observe the same pattern for more frequent cell populations, as clearly stated:

Line 241: *"All IFN- γ (effect range 0.12 to 0.22) and IL-2⁺ (effect range 0.13 to 0.21) as well as double cytokine producing TNF- α IL-2⁺ (effect range 0.13 to 0.22) and polyfunctional (effect range 0.11 to 0.22) CD4⁺ T-cell responses against all four proteins increased with each unit of disease severity (Figure 3C and 3D and Table S2)"*

Further, we have skipped the word polyfunctional from the heading in line 107 “All four SARS-CoV-2 proteins induced ~~polyfunctional~~ CD4⁺ T-cell responses in cases”

In addition, after carefully looking at the automatic gating strategy exemplarily shown in Suppl. S13, gates cut through the negative population, leading to high frequencies of false positive cells (e.g. 1.09% of IFN γ single positive CD4⁺ cells from which the majority is false positive). Gates look way different in Fig. 1 and 2, leaving the reviewer insecure how specific T cells were quantified overall.

We apologize for this oversight, the Suppl S13 was based on our earliest gating strategies that following careful correlation with manual gating strategy was changed to what can be observed in Fig. 1 and 2. We have now updated Suppl S13 to show the same individual as is shown in Figure 1B and 2B upon S stimulation, with the correct gating strategy. The approach used in the manuscript, adding 0.1 offset (log10 scale) to median cytokine levels (see Methods for Automatic gating subsection 5) is clarified in the updated S13 by marking the median cytokine level with a red cross and the cutoff for positive cytokine is labeled with black lines.

Figure S13 updated